# Inhibition of Notch pathway arrests PTEN-deficient advanced prostate cancer by triggering p27-driven cellular senescence

Ajinkya Revandkar[1,2], Maria Luna Perciato[1,*], Alberto Toso[1,*], Abdullah Alajati[1], Jingjing Chen[1,2], Hermeto Gerber[3,4,5], Mitko Dimitrov[3], Andrea Rinaldi[1], Nicolas Delaleu[6], Emiliano Pasquini[1], Rocco D'Antuono[7], Sandra Pinton[1], Marco Losa[1], Letizia Gnetti[8], Alberto Arribas[1], Patrick Fraering[3,4,5], Francesco Bertoni[1], Alain Nepveu[9] & Andrea Alimonti[1,2]

Activation of NOTCH signalling is associated with advanced prostate cancer and treatment resistance in prostate cancer patients. However, the mechanism that drives NOTCH activation in prostate cancer remains still elusive. Moreover, preclinical evidence of the therapeutic efficacy of NOTCH inhibitors in prostate cancer is lacking. Here, we provide evidence that PTEN loss in prostate tumours upregulates the expression of *ADAM17*, thereby activating NOTCH signalling. Using prostate conditional inactivation of both *Pten* and *Notch1* along with preclinical trials carried out in *Pten*-null prostate conditional mouse models, we demonstrate that *Pten*-deficient prostate tumours are addicted to the NOTCH signalling. Importantly, we find that pharmacological inhibition of γ-secretase promotes growth arrest in both *Pten*-null and *Pten/Trp53*-null prostate tumours by triggering cellular senescence. Altogether, our findings describe a novel pro-tumorigenic network that links PTEN loss to ADAM17 and NOTCH signalling, thus providing the rational for the use of γ-secretase inhibitors in advanced prostate cancer patients.

[1] Institute of Oncology Research (IOR) and Oncology Institute of Southern Switzerland (IOSI), Bellinzona CH 6500, Switzerland. [2] Faculty of Biology and Medicine, University of Lausanne (UNIL), Lausanne CH 1011, Switzerland. [3] Brain Mind Institute and School of Life Sciences, Ecole Polytechnique Fédérale de Lausanne (EPFL), Lausanne CH 1015, Switzerland. [4] Foundation Eclosion, Plan-Les-Ouates CH 1228, Switzerland. [5] Campus Biotech Innovation Park, Geneva CH 1202, Switzerland. [6] Broegelmann Research Laboratory, Department of Clinical Science, University of Bergen, Bergen 5021, Norway. [7] Institute for Research in Biomedicine, University of Italian Switzerland, Via Vincenzo Vela 6, Bellinzona 6500, Switzerland. [8] Pathology Unit, University Hospital of Parma, Parma 43126, Italy. [9] Rosalind and Morris Goodman Cancer Research Center, Department of Oncology, Biochemistry and Medicine, McGill University, Montreal, Quebec, Canada H3A1A3. * These authors contributed equally to this work. Correspondence and requests for materials should be addressed to A.A. (email: andrea.alimonti@ior.iosi.ch).

Prostate cancer (PCa) is the most commonly diagnosed tumour type in men and the second leading cause of cancer-related mortality in the United States of America[1,2]. In PCa, enhanced NOTCH activation has been associated with tumour initiation, progression and treatment resistance[3–14]. These findings, together with the recent demonstration that docetaxel-resistant PCa cells upregulate NOTCH signalling, have generated interest for the use of NOTCH inhibitors in PCa patients[11]. Activating mutations of *NOTCH1* receptor are frequently found in leukemia and lung cancer[15–18]. However, *NOTCH* mutations in PCa have been infrequently reported, therefore, how NOTCH signalling is regulated in PCa still remains elusive. A better understanding of the molecular mechanism that drives activation of NOTCH signalling in PCa would be of great relevance for the clinical development of NOTCH inhibitors for PCa patients.

Activation of NOTCH, initiated either by ligand-receptor interactions or due to mutations in NOTCH receptor, requires two consecutive proteolytic cleavages of the NOTCH receptors[19]. The first cleavage at site S2 is mediated by two members of the family of a disintegrin and metalloproteases (ADAMs), ADAM10 or ADAM17 (ref. 20), the second at site S3 by the γ-secretase complex[21]. These events generate the release of NOTCH intracellular domain (NICD), which translocates into the nucleus to regulate the transcription of NOTCH target genes[22]. ADAMs are membrane-associated metalloproteases that possess a complex multi-domain structure, with more than half of the members exhibiting proteolytic potential. Although ADAM10 mediate S2 cleavage in a ligand-dependent manner, ADAM17 cleaves Notch in the absence of ligand, a process may be important in tumours overexpressing ADAM17 protein. Interestingly, ADAMs are upregulated in a number of cancers, including PCa[23]. In an attempt to identify the regulators of NOTCH signalling in PCa, we analysed available gene expression profile data sets[24] from different *Pten*-deficient prostate conditional mouse models wherein we found that several NOTCH target genes were strongly upregulated in these tumours (Supplementary Fig. 1a). *PTEN* gene alterations account for nearly 40% of PCa cases wherein it is also responsible for treatment resistance[25–28]. Moreover, *Pten* is a haplo-insufficient tumour-suppressor gene and variation in Pten protein levels promote PCa in mice in the absence of *Pten* genetic alterations[29,30]. Previous evidence demonstrates that NOTCH1 activation can repress PTEN levels initiating tumorigenesis[31]. However, whether PTEN loss could activate NOTCH signalling remains unknown. Here we have put forward the hypothesis that PTEN loss may trigger NOTCH signalling by regulating the NOTCH proteolytic cleavage.

Our results show that, in PCa, loss of PTEN enhances the levels of ADAM17 thereby promoting the activation of NOTCH signalling. This PTEN/NOTCH axis is sustained by the oncogenic isoform of the transcription factor (TF) CUX1 (Cut-Like Homeobox 1) (p110 CUX1), a previously unknown regulator of *ADAM17* transcription. The p110 CUX1 protein is a proteolytic isoform of the full-length p200 CUX1, cleaved by CathepsinL[32]. Recent findings demonstrate that increased expression of p110 CUX1 functions as a transcriptional activator of genes involved in tumour cell proliferation and invasiveness[33–35]. Importantly, we demonstrate that treatment with a γ-secretase inhibitor (GSI) is highly effective in blocking PCa progression in different *Pten*-deficient mouse models of PCa.

## Results

**Loss of Pten activates Notch signalling in PCa.** To study whether loss of PTEN impacts NOTCH signalling in prostate tumours, we took advantage of the *Pten* prostate conditional mouse model (here after referred to as *Pten*[pc −/−])[36]. Intracellular proteolytic processing of the NOTCH1 receptor leads to the generation of NICD1, a marker of NOTCH1 activation[37]. Immunohistochemical (IHC) and western blot (WB) analyses for NICD1 showed that Notch signalling is highly activated in *Pten*-deficient prostate tumours (Fig. 1a–c). In line with these observations, Gene Set Enrichment Analysis (GSEA) of *Pten*[pc −/−] mouse tumours revealed a Notch signature in *Pten*[pc −/−] tumours (Fig. 1d). These data were also validated in a human PCa data set where *PTEN* mRNA levels inversely correlated with Hairy and enhancer of split1 (*HES1*) levels in PCa patients at different stages of disease (Supplementary Fig. 1b). In line with increased levels of NOTCH signature observed in *Pten*[pc −/−] tumours, enzymatic activity assay confirmed that the γ-secretase complex was activated to a higher extent in *Pten*[pc −/−] tumours as compared with control prostates (Supplementary Fig. 1c,d).

To evaluate the relevance of Notch1 signalling in *Pten*[pc −/−] tumours, we generated prostate conditional knockout of both *Pten* and *Notch1* in mice (hereafter referred as *Pten*[pc −/−]; *Notch1*[pc −/−]). We first confirmed the prostate-specific deletion of both *Pten* and *Notch1* by WB analysis and reverse transcription PCR (RT-PCR) for Notch target genes (Supplementary Fig. 1e). Next, we performed histopathological analysis to determine the effect of combined *Pten* and *Notch1* inactivation in mouse prostates. Mice were killed at different ages and prostates were resected from *Pten*[pc +/+], *Notch1*[pc −/−], *Pten*[pc −/−] and *Pten*[pc −/−]; *Notch1*[pc −/−] mice. Notably, inactivation of *Notch1* in *Pten*[pc −/−] tumours induced a strong inhibition of tumour growth compared with age-matched *Pten*[pc −/−] littermates as measured by decreased number of glands affected by high-grade prostatic intraepithelial neoplasia and invasive PCa (Fig. 1e,f and Supplementary Fig. 1f). Furthermore, *Pten*[pc −/−]; *Notch1*[pc −/−] tumours showed reduced cell proliferation as observed by a decrease in Ki-67 staining compared with *Pten*[pc −/−] tumours (Fig. 1g and Supplementary Fig. 1f). Note that both *Pten*[pc +/+] and *Notch1*[pc −/−] mice did not develop any prostate lesions. Interestingly, the reduction of cell proliferation in *Pten*[pc −/−]; *Notch1*[pc −/−] tumours was accompanied by a strong increase in p27 protein levels and upregulation of the senescence-associated beta-galactosidase staining, two markers of senescence (Fig. 1h,i and Supplementary Fig. 1g). The enhanced γ-secretase activity in *Pten*[pc −/−] tumours and their dependence on Notch1 signalling suggested that *Pten*-deficient prostate tumours could be highly responsive to the treatment with GSIs.

**PF-03084014 promotes senescence in advanced PCa.** Having demonstrated the relevance of activated Notch1 signalling in *Pten*[pc −/−] prostate tumorigenesis, we conducted a preclinical trial to test the impact of γ-secretase inhibition in *Pten*[pc −/−] mice. We therefore treated a cohort of *Pten*[pc −/−] mice with PF-03084014, a GSI currently in clinical evaluation[38,39]. Treatment was started when the prostate tumours were already established at 8 weeks of age (Fig. 2a). At the end of the treatment, mice were killed and prostate tumours were isolated for analysis. Treatment with PF-03084014 was well tolerated, as indicated by the absence of body weight loss (Supplementary Fig. 2a), and mice did not develop any toxicities with the exception of a spotted loss of hair pigmentation phenotype, an indicator of the activity of the compound *in vivo* (Supplementary Fig. 2b)[40,41]. Gross anatomy of different prostate lobes revealed that in *Pten*[pc −/−] mice treated with PF-03084014, the tumour size was strongly reduced when compared with control tumours (Fig. 2b,c). Reduction in tumour size was associated with an enhanced senescence response as measured by decreased Ki-67 staining, increased pHP1γ

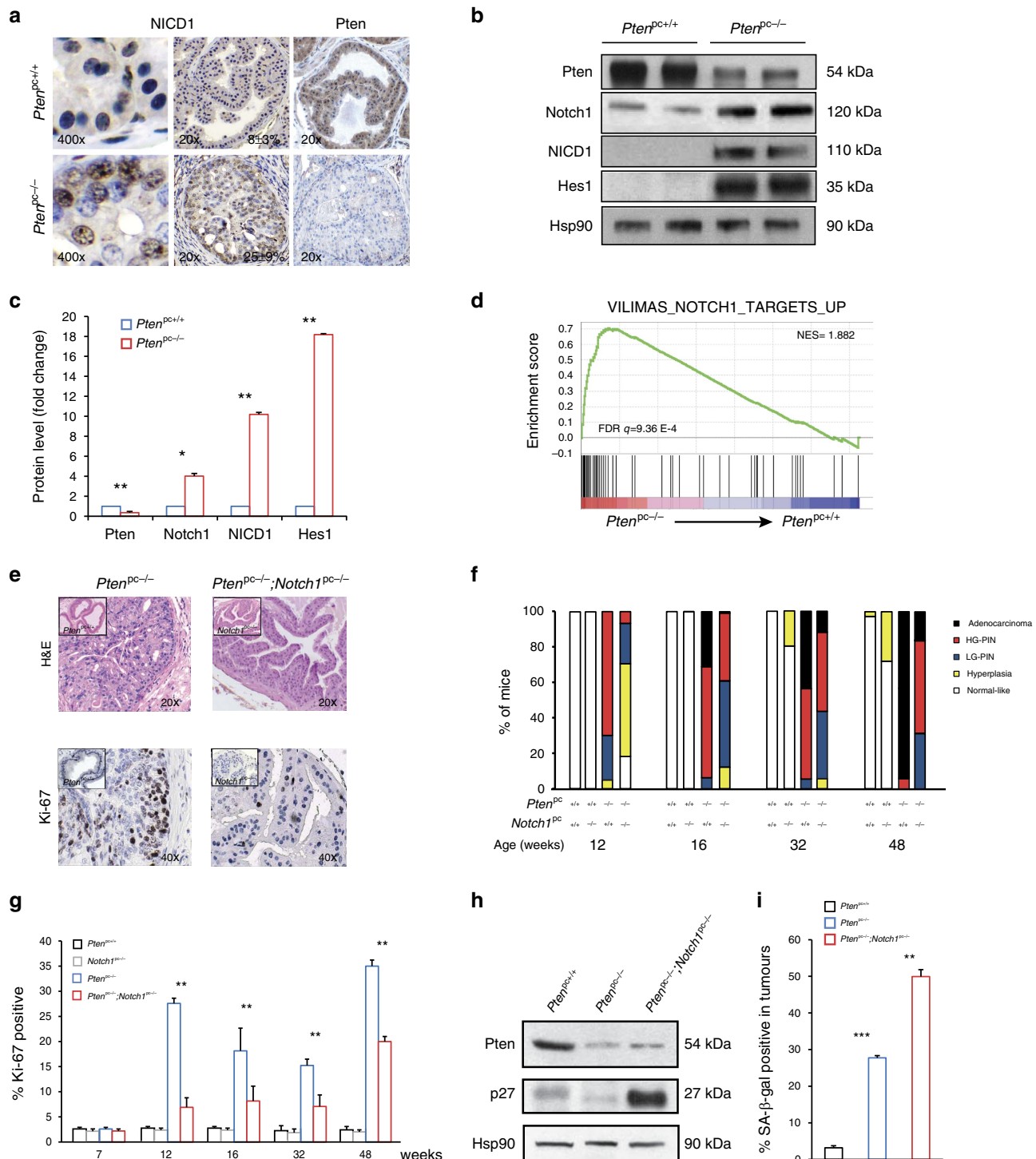

**Figure 1 | Notch1 signalling is activated in *Pten*^pc −/− prostate tumours.** (**a**) IHC showing NICD1 and Pten stainings in both *Pten*^pc+/+ normal prostate (8 ± 3%) and *Pten*^pc −/− prostatic intraepithelial neoplasia (PIN) lesions resected from 12-week-old mice (25 ± 9%) (*n* = 3). Magnification × 20 and × 400. (**b**) WB showing the protein levels of Pten, total Notch1, NICD1 and Hes1 in both *Pten*^pc+/+ prostate and *Pten*^pc −/− prostate tumours (*n* = 5). (**c**) Quantification of **b**. (**d**) Gene Set Enrichment Analysis (GSEA) showing activation of Notch1 signalling in *Pten*^pc −/− prostate tumours. (**e**) H&E and Ki-67 staining of APs derived from 12-week-old *Pten*^pc −/− and *Pten*^pc −/− ; *Notch1*^pc −/− tumours (*n* = 5). Magnification × 20 and × 40, respectively. Insets represent H&E and Ki-67 images of *Pten*^pc+/+ and *Notch1*^pc −/− . (**f**) Histopathological characterization of normal prostates and prostate tumours in mice of the indicated genotypes (*n* = 16 for each genotype). (**g**) Quantification of Ki-67 staining of APs in mice of indicated genotypes. (**h**) WB showing the protein levels of Pten and p27 in *Pten*^pc+/+, *Pten*^pc −/− and *Pten*^pc −/− ; *Notch1*^pc −/− prostate tumours. (**i**) Quantification of the percentage of Senescence-associated beta-galactosidase staining in *Pten*^pc+/+, *Pten*^pc −/− and *Pten*^pc −/− ; *Notch1*^pc −/− prostate tumours. Values are expressed as mean ± s.e.m. *P < 0.05; **P < 0.01; ***P < 0.001 by Student's *t*-test.

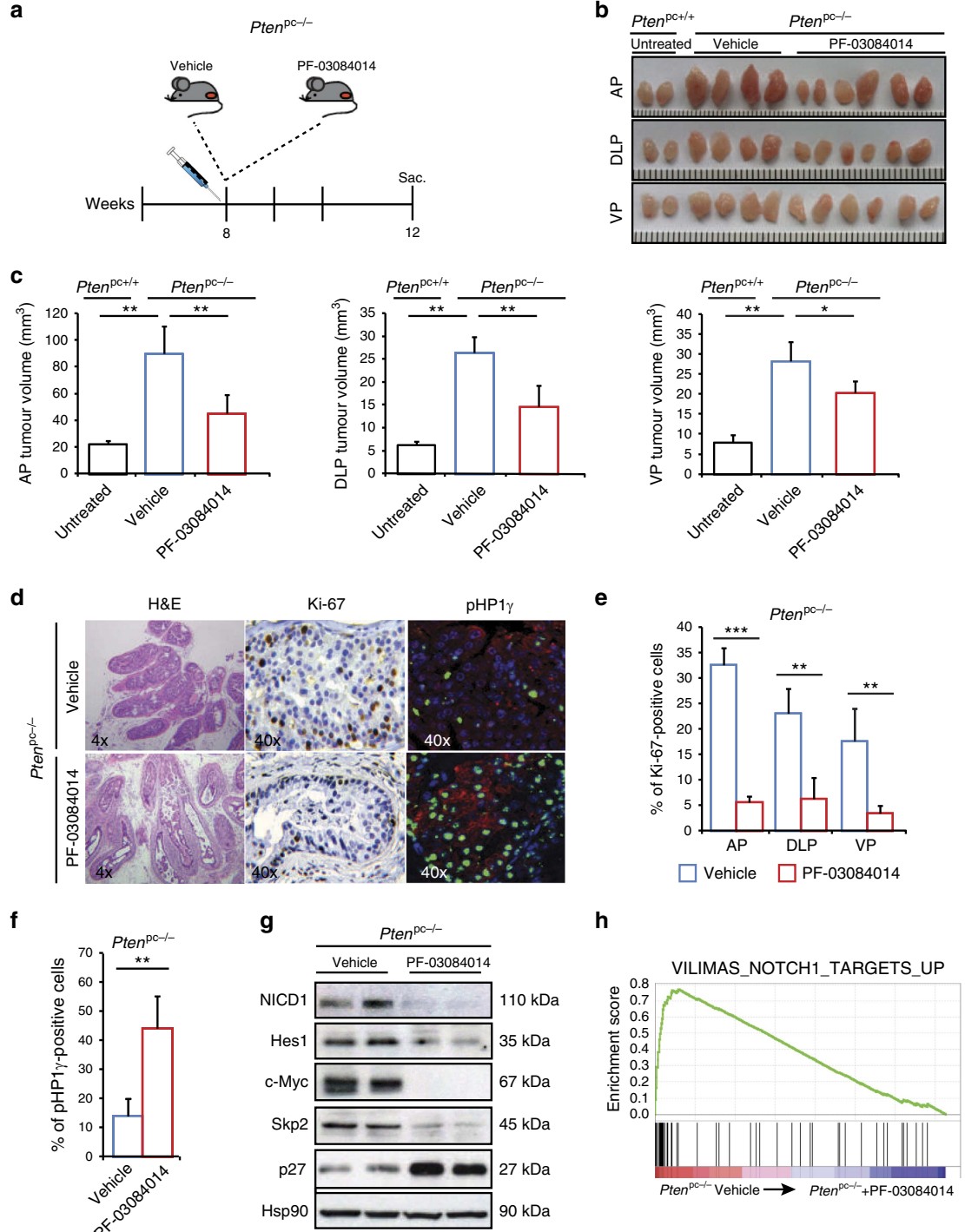

**Figure 2 | PF-03084014 constrains tumorigenesis of Pten$^{pc−/−}$ prostate tumours. (a)** Scheme of treatment. Mice have been treated twice a day for 4 weeks with either vehicle or PF-03084014 (100 mg kg$^{−1}$). **(b)** Picture showing the size of anterior (AP), dorso-lateral (DLP) and ventral prostates (VP) from Pten$^{pc+/+}$ wt prostates and Pten$^{pc−/−}$ tumours treated with either vehicle or PF-03084014. **(c)** Quantification of **b** ($n = 5$). **(d)** H&E, Ki-67 and pHP1γ staining of APs derived from Pten$^{pc−/−}$ tumours treated with either vehicle or PF-03084014. Magnification × 4 and × 40. **(e,f)** Quantification of **d** ($n = 5$). **(g)** WB on AP extracts Pten$^{pc−/−}$ tumours showing the effect of PF-03084014 on Notch1 signalling. **(h)** Gene Set Enrichment Analysis (GSEA) showing reduced Notch1 signalling in Pten$^{pc−/−}$ prostate tumours treated with PF-03084014. Values are expressed as mean ± s.e.m. *$P < 0.05$; **$P < 0.01$; ***$P < 0.001$ by Student's $t$-test.

staining (Fig. 2d–f) and p27 protein levels (Fig. 2g). Interestingly, upon γ-secretase inhibition we found reduced levels of S-phase kinase-associated protein 2 (Skp2), an inhibitor of p27, that was previously shown to regulate senescence in Pten$^{pc−/−}$ tumours both by mRNA and protein levels[42]. Reduced levels of NICD1 and Notch1 target genes, such as Skp2, c-Myc and *Hes1*,

confirmed that PF-03084014 efficiently reached the target in Pten$^{pc−/−}$ tumours (Fig. 2g). GSEA analysis further confirmed loss of Notch1 signalling in Pten$^{pc−/−}$ tumours treated with PF-03084014 (Fig. 2h). Upregulation of senescence and p27 was also validated in Pten-null mouse embryonic fibroblasts (MEFs) treated with PF-03084014 (Supplementary Fig. 2c,d). Notably,

treatment of $Pten^{pc-/-}$ tumours with PF-03084104 blocked the expression of *Notch1* as confirmed by RT-PCR (Supplementary Fig. 2e). This data suggest that Notch1 activation induces its own expression, in line with previously published results[43-46]. Guided by the results obtained in $Pten^{pc-/-}$ mice, we also assessed the efficacy of PF-03084014 in more aggressive mouse models of PCa. $Pten^{pc-/-}$ mice from 15 weeks of age and $Pten^{pc-/-}$; $Trp53^{pc-/-}$ develop focal invasive and fully invasive PCa, respectively. In particular, $Pten^{pc-/-}$; $Trp53^{pc-/-}$ tumours are considered highly aggressive and resistant to the majority of conventional treatments available in the clinic for PCa[36]. Treatment was started when the prostate tumours were invasive (15 weeks of age) and lasted for 5 weeks. $Pten^{pc+/+}$ and $Trp53^{pc-/-}$ mice also included in this trial and used as a control to assess potential negative effects of PF-03084014 treat-

ment in normal-like prostate tissues. Remarkably, haematoxylin and eosin and immunofluorescence (IF) staining for Vimentin/ E-Cadherin revealed that, in $Pten^{pc-/-}$; $Trp53^{pc-/-}$ mice treated with PF-03084014, tumour grade and percentage of invasive glands was significantly reduced when compared with $Pten^{pc-/-}$; $Trp53^{pc-/-}$ control mice (Fig. 3a). The reduced invasiveness was accompanied by decreased proliferation as measured by Ki-67 staining and reactive tumour stroma (Fig. 3a). We also observed decreased invasiveness and proliferation in 15-week-old $Pten^{pc-/-}$ tumours upon treatment with PF-03084014, validating the results obtained with mice at early age (Supplementary Fig. 3a,b). As observed in $Pten^{pc-/-}$ mice, treatment with PF-03084014 increased p27 levels and decreased Skp2 protein levels also in $Pten^{pc-/-}$; $Trp53^{pc-/-}$ tumours (Fig. 3b–d). However, treatment with PF-03084014 did not

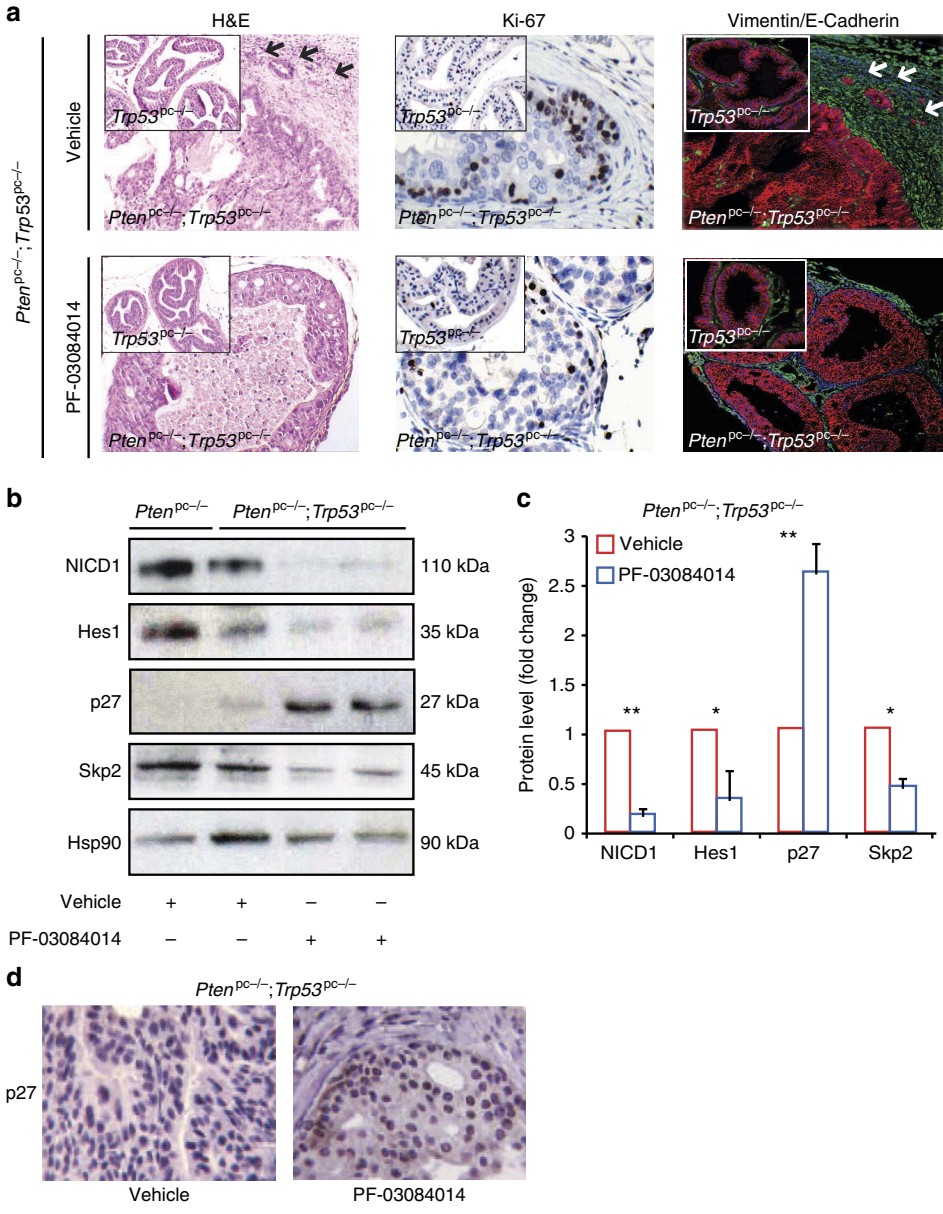

**Figure 3 | Anti-tumour activity of PF-03084014 in $Pten^{pc-/-}$;$Trp53^{pc-/-}$ prostate tumours.** (**a**) Representative H&E, Ki-67 and Vimentin/E-Cadherin immunofluorescence stainings of $Pten^{pc-/-}$; $Trp53^{pc-/-}$ prostate tumours treated with either vehicle or PF-03084014. White arrow shows an invasive area of epithelial tumour cells infiltrating the tumour stroma. Insets represent H&E, Ki-67 and Vimentin/E-Cadherin staining in $Trp53^{pc-/-}$ normal prostates treated with either vehicle or PF-03084014. (**b**) WB on AP extracts from $Pten^{pc-/-}$; $Trp53^{pc-/-}$ tumours showing the effect of PF-03084014 on Notch1 signalling. (**c**) Quantification of **b**. (**d**) p27 staining in $Pten^{pc-/-}$; $Trp53^{pc-/-}$ tumours treated with vehicle and PF-03084014. Values are expressed as mean ± s.e.m. *$P < 0.05$; **$P < 0.01$ by Student's t-test.

change the morphology and proliferation of normal-like prostates in both $Pten^{pc+/+}$ and $Trp53^{pc-/-}$ mice (Supplementary Fig. 3a,b). Finally, treatment of LNCaP and PC3, two *PTEN*-deficient human PCa cell lines, with PF-03084014 also showed significant cell growth inhibition upon Notch pathway inhibition in agreement with our findings in mice and recent evidence[47] (Supplementary Fig. 3c,d). Taken together, these data demonstrate that NOTCH inhibition is effective in blocking *Pten* loss driven tumorigenesis at both early and late stage.

**PTEN loss triggers ADAM17 upregulation in PCa.** Activation of the NOTCH1 receptor is a tightly regulated process that relies on a hierarchically ordered proteolytic cleavage cascade[48]. Generation of intracellular NICD1 by the γ-secretase complex follows, and requires, the initial extracellular cleavage of NOTCH1 by ADAM metalloproteases[49]. We therefore checked whether loss of *Pten* could alter the expression of Adam17, a metalloprotease that leads to NOTCH activation independently of the presence of the NOTCH ligand. We found that Adam17 was upregulated at both the protein and mRNA levels in $Pten^{pc-/-}$ and protein level in $Pten^{pc-/-}$; $Trp53^{pc-/-}$ mice (Fig. 4a–c and Supplementary Fig. 4a). Similarly, ADAM17 was specifically upregulated in human PCa cell lines having functional loss of PTEN in either one (DU-145) or both (LNCaP, PC3) alleles. This was also associated with increased levels of NICD1 and Hairy/enhancer-of-split related with YRPW motif protein 1 (*HEY1*) (Fig. 4d–f and Supplementary Fig. 4b).

Consistent with our data obtained in $Pten^{pc-/-}$ mice, genetic inactivation of *PTEN* in DU-145 cells led to an increased expression of ADAM17 along with NICD1 upregulation (Supplementary Fig. 4c,d). We next checked the correlation between the protein levels of PTEN and ADAM17 in two different tissue microarrays ($n=130$) of human PCa. Our analysis revealed that the majority of samples displaying low levels of PTEN stained positive for ADAM17 (Fig. 4g,i; $P=0.039511$). Bioinformatics analysis also showed an inverse correlation between the gene expression levels of *PTEN* and *ADAM17* in prostate tumours (Fig. 4h). Low levels of PTEN and high levels of ADAM17 in tumours correlated with high tumour grading ($P<0.0001$) and Gleason score ($P<0.0001$) (Fig. 4i). Patients with tumours characterized by low levels of *PTEN* and high levels of *ADAM17* ($PTEN^{low}ADAM17^{high}$) had also a worse clinical outcome compared with other patient groups (Supplementary Fig. 4e). Altogether, these data indicate that PTEN loss drives the upregulation of ADAM17 and the consequent activation of NOTCH pathway. Importantly, inhibition of ADAM17 by means of a short hairpin RNA (shRNA) also decreased NICD1 protein level and reduced the proliferation of the aggressive human PCa cell line PC3 (Fig. 4j,k).

**CUX1 regulates the levels of ADAM17 in prostate tumours.** Our data show that loss of PTEN leads to increased levels of ADAM17. To assess whether this effect was phosphatidylinositol 3′-kinase (PI3K)/AKT dependent, we treated human PCa cell lines with either normal or functional loss of PTEN with the PI3K inhibitor, AZD8186. Interestingly, inhibition of PI3K/AKT in these cells did not affect ADAM17 levels, suggesting that activation of NOTCH signalling is PI3K/AKT-independent (Supplementary Fig. 5). We therefore looked for a TF that could regulate *ADAM17* expression in PTEN-deficient tumour cells independently of the PI3K/AKT pathway. To this end, we screened in the SABiosciences' proprietary database (DECODE, DECipherment Of DNA Elements) for TFs that were predicted to bind the *ADAM17* promoter in both mouse and human. Among the predicted TFs identified, CUX1 TF was the only conserved TF predicted to bind to the promoter of both mouse and human

*ADAM17* (Supplementary Table 1). Given that PTEN loss enhances ADAM17 levels in both mouse and human PCa cells, we focussed on this TF. Interestingly, evidence exists that p110 CUX1, a cleaved form of full-length CUX1 (p200 CUX1), has a potential oncogenic function[35]. Indeed, transgenic mouse model overexpressing p110 CUX1 develop mammary carcinomas[33]. We then checked the expression levels of CUX1 in $Pten^{pc-/-}$ tumours. Surprisingly, WB analysis revealed that p110 CUX1 was overexpressed in $Pten^{pc-/-}$ tumours, while p200 CUX1 was mainly expressed in $Pten^{pc+/+}$ prostates (Fig. 5a,b). The p110 CUX1 isoform is a result of a proteolytic cleavage of the full-length p200 CUX1 mediated by CathepsinL (Fig. 5a). Consistent with the high levels of p110 CUX1, we found that CathepsinL was strongly overexpressed in $Pten^{pc-/-}$ tumours (Fig. 5a,b). These results were further confirmed in $Pten^{-/-}$ MEFs by both WB and IF analyses (Fig. 5c,d and Supplementary Fig. 6a). It is important to note that IF staining also showed an accumulation of CUX1 in the nucleus of $Pten^{-/-}$ MEFs (Fig. 5d and Supplementary Fig. 6b). Importantly, high levels of p110 CUX1 were also observed in all the PCa cells analysed, whereas p200 CUX1 was barely detected (Fig. 5f). Next, we performed chromatin immunoprecipitation (ChIP) experiment in PC3 to assess the binding of CUX1 on the promoter of *ADAM17*. In agreement with the CUX1-binding site prediction, we found that CUX1 strongly bound the promoter of *ADAM17* (Fig. 5g). Furthermore, we knocked down *CUX1* in PC3 by means of two different small interfering RNAs (siRNAs). Strikingly, downregulation of CUX1 in PC3 strongly reduced ADAM17 at both the gene expression and protein levels (Fig. 5h,i). Similar results for ChIP and siRNA experiments were obtained in DU-145 PCa cells (Supplementary Fig. 6c,d). Consistent with the siRNA experiments, treatment of PC3 with a CathepsinL inhibitor[32], which blocks the conversion of p200 CUX1 in p110 CUX1, decreased the levels of both ADAM17 and NICD1 in PC3 cells (Supplementary Fig. 6e).

**p110 CUX1 upregulates ADAM17/NOTCH signalling.** Finally, we investigated whether overexpresion of p110 CUX1 resulted in activation of NOTCH signalling. Transient overexpression of p110 CUX1 in the PCa cell line 22Rv1 increased both the ADAM17 and NICD1 protein levels (Supplementary Fig. 7a,b). Overexpression of p110 CUX1 in the same cells also increased the luciferase activity of ADAM17 promoter by nearly 10-fold when compared with the empty vector (Supplementary Fig. 7c). Previous findings showed that overexpression of p110 CUX1 in the mouse mammary gland led to the formation of invasive breast cancer[33]. IHC and WB analyses in the mammary glands of p110 CUX1 transgenic mice showed an enhanced protein expression of both Adam17 and NICD1 as compared with control mammary glands (Supplementary Fig. 7d–f). p110 CUX1 overexpression in the breast carcinoma cell line Hs578T also increased the expression of *ADAM17* and *HEY1*, a downstream target of NOTCH (Supplementary Table 2)[50], whereas downregulation of p110 CUX1 in the same cells decreased the NOTCH signalling. In sum, our findings demonstrate that p110 CUX1 enhanced the levels of ADAM17, which in turn leads to NOTCH signalling activation validating our observation *in vivo* in different tumour models.

**Discussion**
Activated NOTCH signalling has been reported to be associated with advanced[12,13] and metastatic PCa[4,8,14]. However, the mechanism behind NOTCH activation in prostate tumours remained elusive so far. To address this question, we used $Pten^{pc-/-}$ and $Pten^{pc-/-}$; $Trp53^{pc-/-}$ mouse models that

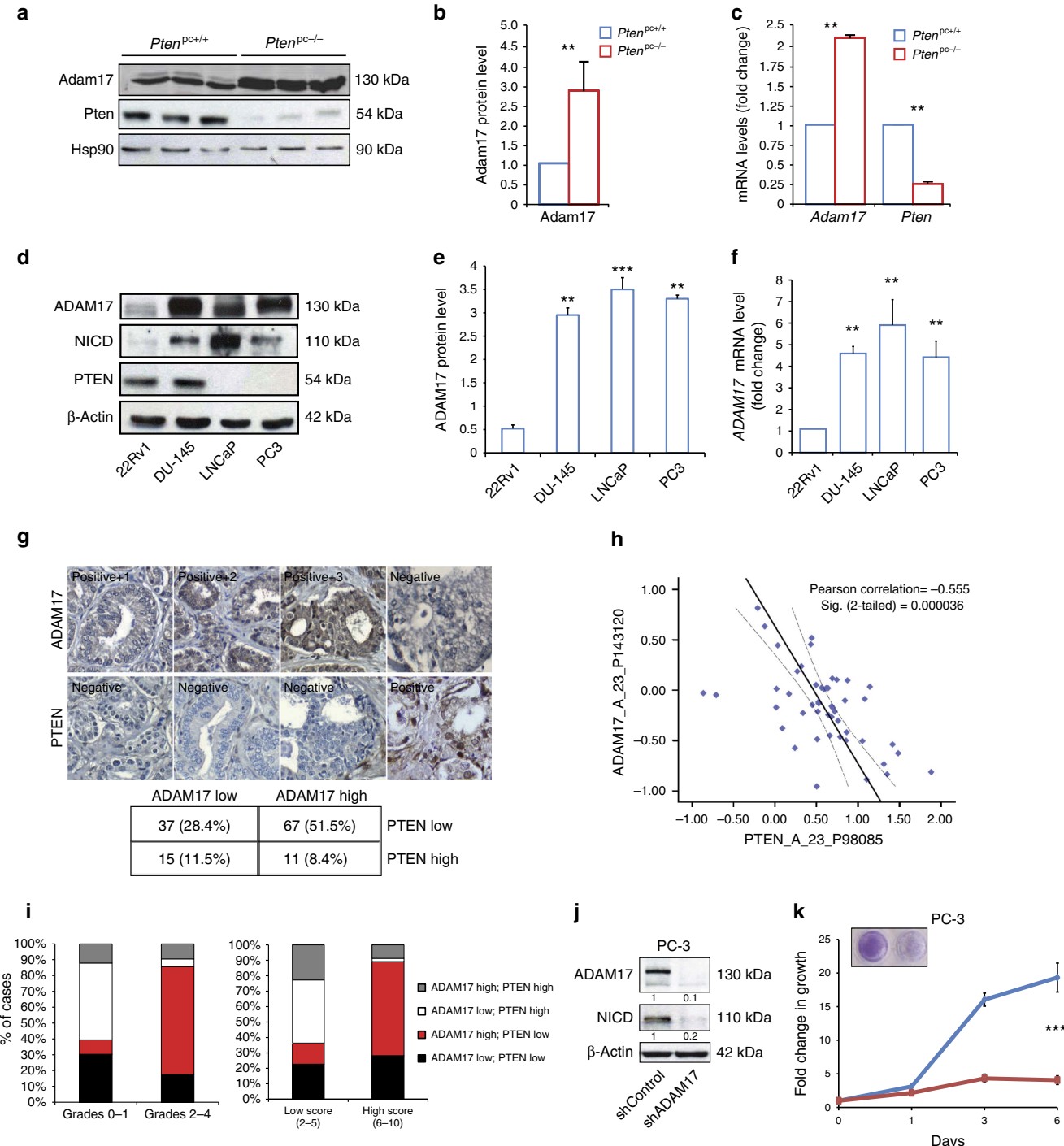

**Figure 4 | ADAM17 is upregulated in PCa.** (**a**) WB showing the protein levels of Adam17 in $Pten^{pc-/-}$ tumours compared with $Pten^{pc+/+}$ *WT* prostates. (**b**) Quantification of **a** ($n = 3$). (**c**) *Adam17* and *Pten* mRNA levels in both $Pten^{pc+/+}$ prostates and $Pten^{pc-/-}$ tumours. (**d**) WB showing the protein levels of ADAM17, NICD and PTEN in different PCa cell lines. (**e**) Quantification of ADAM17 of **d**. (**f**) *ADAM17* mRNA levels in different PCa cell lines. (**g**) ADAM17 and PTEN staining on human prostate cancer tissue microarray (TMA). Table showing correlation between ADAM17 and PTEN staining quantification. Data of two different TMA were combined (total no. of samples = 130). (**h**) Inverse correlation between the mRNA levels of *PTEN* and *ADAM17* in human prostate cancers. (**i**) Bar graphs representing the correlation of ADAM17 and PTEN levels with tumour grade and Gleason score. (**j**) WB for ADAM17 and NICD1 in PC3 cells infected with either an *shRNA* control or *shADAM17*. (**k**) Growth curve of PC3 cells infected with either an *shRNA* control or *shADAM17*. Values are expressed as mean ± s.e.m. \*\**P* < 0.01; \*\*\**P* < 0.001 by Student's *t*-test.

develop high-grade prostatic intraepithelial neoplasia and invasive prostate tumours, respectively[36]. In line with human data, we observed an increased NOTCH signalling in these mouse models as documented by the increased γ-secretase activity and upregulation of different NOTCH-targeted genes (e.g., *Hes1*,

*Ccnd1*). Furthermore, to determine the role of Notch signalling in *Pten* loss-driven prostate tumorigenesis, we generated combined conditional inactivation of *Pten* and *Notch1* in mouse prostatic epithelia. Strikingly, genetic inactivation of *Notch1* in $Pten^{pc-/-}$ mice almost abrogated prostate tumorigenesis by strongly

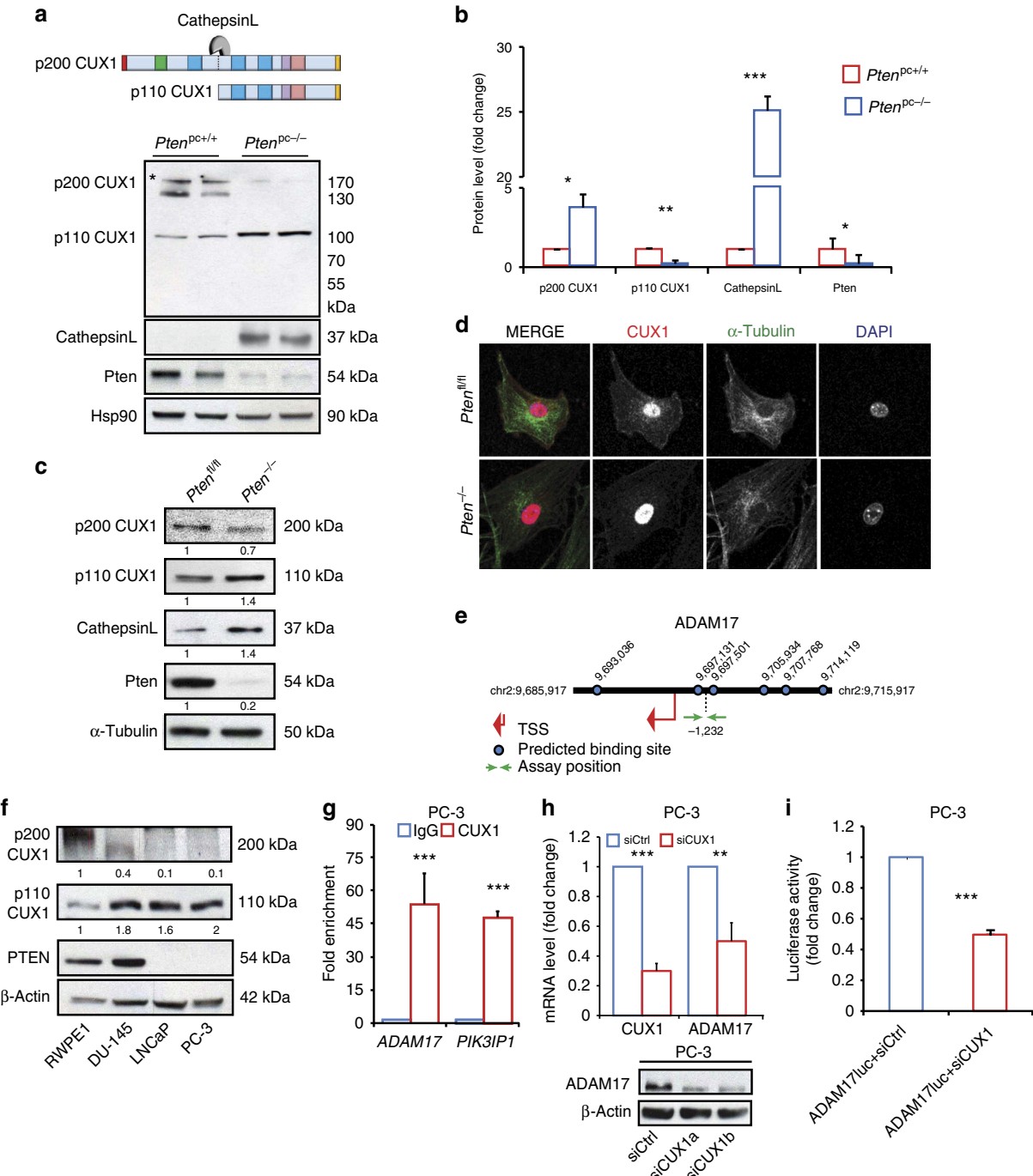

**Figure 5 | CUX1 regulates ADAM17 expression in prostate tumour.** (**a**) Schematic representation of the full-length CUX1 (p200 CUX1) and its oncogenic isoform (p110 CUX1). WB for p200 Cux1, p110 Cux1, Pten and CathepsinL in both $Pten^{pc+/+}$ prostates and $Pten^{pc-/-}$ tumours. (**b**) Quantification of **a** ($n = 3$–5). (**c**) WB showing the levels of Cux1, CathepsinL and Pten in both $Pten^{fl/fl}$ and $Pten^{-/-}$ MEFs. (**d**) IF images showing the localization of Cux1 in both $Pten^{fl/fl}$ and $Pten^{-/-}$ MEFs. (**e**) Schematic representation of $ADAM17$. The blue dots indicate the CUX1 predicted binding sites. In green the region where the primers used for the ChIP experiments have been designed. (**f**) CUX1 and PTEN in different PCa cell lines. (**g**) CUX1 ChIP. Graph showing the fold enrichment of $ADAM17$ and $PIK3IP1$ (CUX1 target) genes in PC3 cells. (**h**) mRNA levels and WB for CUX1 and ADAM17 in PC3 cells. (**i**) ADAM17 luciferase activity upon knockdown of CUX1. Values are expressed as mean ± s.e.m. $*P < 0.05$; $**P < 0.01$; $***P < 0.001$ by Student's $t$-test.

reducing cell proliferation. This result demonstrates that prostate tumours driven by loss of $Pten$ require the activation of Notch1. Notably, we observed that both genetic and pharmacological inhibition of NOTCH signalling were associated with inhibition of proliferation and upregulation of a p27-mediated cellular senescence response in both $Pten^{pc-/-}$ and $Pten^{pc-/-}$; $Trp53^{pc-/-}$ tumours. As previously documented, tumours

affected by the combined loss of $Pten$ and $Trp53$ are insensitive to the majority of therapies available in the clinic for PCa, such as androgen deprivation and docetaxel, and it is therefore surprising to observe that PF-03084014 is highly effective in this tumour background[51]. Activation of p27 in $Pten$-null tumours was associated with decreased Skp2 levels. SKP2 is a known NOTCH target gene and it is a regulator of p27 degradation.

Treatment with the GSI, PF-03084014, in both $Pten^{pc-/-}$ and $Pten^{pc-/-}$; $Trp53^{pc-/-}$ mice strongly decreased the protein levels of NICD1 and Hes1, thus confirming that the GSI reached its target. These findings are in line with a previous study demonstrating that treatment with a Skp2 inhibitor in a PC3 xenograft mouse model (PC3 cells lack both PTEN and p53) blocks tumorigenesis by upregulating both p27 and senescence[42]. However, SKP2 inhibitors have been associated with several side effects in humans and their clinical development has been currently suspended[52–55]. Therefore, it is interesting to note that GSIs work just as well as Skp2 inhibitors but with less toxicity. Our data are also coherent with a recent report demonstrating the efficacy of PF-03084014 in combination with docetaxel in two human prostate xenograft mouse models[47].

Another major advance of our findings is the characterization of the mechanism that links PTEN loss to NOTCH activation in PCa. As upregulation of NOTCH occurred independently of its ligand in PTEN-deficient human PCa cell lines, it remained unclear how aberrant Notch signalling was associated with development and progression of PCa. Analysis of proteases involved in activation of Notch signalling has highlighted an increased S2 processing of Notch receptors mediated by the ADAM17 metalloprotease. ADAM17 is involved in the first cleavage of the NOTCH receptor that results in the generation of the NOTCH extracellular domain. This allows the subsequent cleavage of NOTCH by the γ-secretase complex and translocation of NICD in the nucleus[19,22,49]. Intriguingly, high levels of ADAM17 result in S2 cleavage and subsequent activation of Notch in a ligand-independent manner[15,22,49]. In our mouse models and PTEN-deficient human PCa cell lines, we observed an increase of ADAM17 mRNA levels and an inverse correlation between PTEN and ADAM17, as assessed by histological staining in two different tissue microarrays of human PCa. This was also confirmed by a bioinformatic analysis of additional human PCa data sets. Furthermore, by knocking down ADAM17 in PC3 PCa cells, we found decreased NOTCH1 signalling along with impaired cell proliferation. These data confirmed that NOTCH activation in PTEN-deficient cells is a consequence of ADAM17 upregulation.

Although the regulation of NOTCH1 activity by ADAM17 has been extensively investigated[19,22,23,49], the regulation of ADAM17 itself is a lesser-known phenomenon. Various studies on TFs involved in the upregulation of ADAM17 have shown that under specific conditions, such as hypoxia, ADAM17 expression is upregulated by different TFs[56–58]. In this study, we found that CUX1 was the only TF conserved in both mouse and human predicted to bind to ADAM17. Multiple isoforms of CUX1 have been identified out of which two are ubiquitously expressed. One is the full-length p200 CUX1, known to function as a transcriptional repressor[59], while the other one is a proteolytically cleaved p110 CUX1 isoform, often regarded as transcriptional initiator and found to be overexpressed in multiple cancers[35]. Although p200 CUX1 is known to transiently bind to DNA, the p110 isoform can strongly bind to the promoter region of different genes affecting their transcription[34]. In Pten-null prostate tumours and human cancer cells, we found manifold increase in p110 CUX1 isoform and an undetectable level of p200 CUX1 when compared with the control. In our ChIP analysis, we also found several fold increase in binding of CUX1 to the promoter region of ADAM17 in PCa cell lines, similar to PIK3IP1, a known target of CUX1[60]. Knockdown and overexpression of CUX1 in PCa cells also affected the levels of ADAM17, validating CUX1 as a transcription activator of ADAM17. This has been also found in vivo in a different mouse model where overexpression of p110 CUX1 is associated with NOTCH activation. Collectively, our

observations demonstrate that PTEN loss promotes the activation of NOTCH signalling by upregulating the levels of p110 CUX1 that, in turn, promotes the transcription of ADAM17 (Supplementary Fig. 8). This may happen through an enhanced proteolytic activity of CathepsinL[34], as shown in our study, or additional mechanisms such as unidentified proteases or non-coding RNAs that may lead to either the cleavage or stabilization of p110 CUX1 mRNA. The results presented in this work strengthen the potential therapeutic benefits of targeting γ-secretase in PCa and provide a rationale for stratifying patients who may be more responsive to this treatment owing to loss of PTEN that mediates activation of NOTCH signalling.

## Methods

**Mice.** $Pten^{loxP/loxP}$, $Pten^{loxP/loxP}$; $Notch1^{loxP/loxP}$, $Notch1^{loxP/loxP}$, $Trp53^{loxP/loxP}$ and $Pten^{loxP/loxP}$; $Trp53^{loxP/loxP}$ mice (Jackson Laboratory) were crossed with PB-Cre4 transgenic mice to generate prostate-specific knockout of Pten and Pten; Trp53, respectively[36]. All mice were maintained under specific pathogen-free conditions in the animal facilities of the IRB institute, and the experiments were performed according to the state guidelines and approved by the local ethical committee. GSI PF-03084014, used for all the preclinical trials in a cohort of $Pten^{pc-/-}$ and $Pten^{pc-/-}$; $Trp53^{pc-/-}$ mice (aged 8 and 15 weeks, respectively), was synthesized and provided by Pfizer. Mice undergoing treatment were administered control vehicle or therapeutic doses of PF-03084014 by oral gavage on a monday through friday schedule for a total of 20 days. A dosage of $100\,mg\,kg^{-1}$ of GSI was administered twice-a-day by oral gavage on monday through friday schedule. Mice were monitored for any suffering of distress or weight loss by measuring total body weight of mice biweekly and monitoring the behavioural changes every day for a total of 4 weeks of treatment. Upon completion of study, mice were killed by $CO_2$ asphyxiation, and tissues were procured for histological, mRNA and protein analyses.

**Autopsy and histopathology.** Animals were autopsied, and all tissues were examined regardless of their pathological status. Normal and tumour tissue samples were fixed in 10% neutral-buffered formalin (Sigma) overnight. Tissues were processed by ethanol dehydration and embedded in paraffin according to standard protocols. Sections (5 μm) were prepared for antibody detection and haematoxylin and eosin staining. To evaluate evidence of invasion, sections were cut at 20 μm intervals and haematoxylin and eosin stained. Slides were prepared containing three to five of these interval sections.

**γ-Secretase assay.** γ-Secretase assays using the recombinant human Notch and Amyloid precursor protein substrates Notch100-Flag and APP-C100-Flag were performed as previously reported[61–63]. Membrane proteins were extracted from $Pten^{pc+/+}$- and $Pten^{pc-/-}$-null mice prostate samples in 50 mM HEPES (pH 7.0) with 1% CHAPSO. The protein content in the extracts was normalized by BCA and incubated in 0.2% (wt/vol) CHAPSO, 50 mM HEPES (pH 7.0), 150 mM NaCl, 5 mM $MgCl_2$ and 5 mM $CaCl_2$ and incubated at 37 °C for 4 h with 1 μm substrate, 0.1% (wt/vol) phosphatidylcholine and 0.025% (wt/vol) phosphatidylethanolamine. The generated products AICD (Amyloid Intracellular C-terminal Domain)-Flag and NICD (Notch Intracellular Domain)-Flag were analysed by WB and detected with Flag-specific M2 antibody (Sigma-Aldrich).

**MEF production and cell culture.** Primary MEFs were obtained from individual embryos of $Pten^{loxP/loxP}$ genotype from a pregnant mouse at 13.5 days postcoitum. Primary $Pten^{lox/lox}$ MEFs were infected with retroviruses expressing either pMSCV-CRE-PURO-IRES-GFP or pMSCV-PURO-IRES-GFP for 48 h and selected with Puromycin at a concentration of 3 μg ml$^{-1}$. PTEN wild-type (RWPE-1, 22Rv1) human prostate cell lines and heterozygous or homozygous loss of PTEN function PCa cell lines (DU-145, LNCaP and PC3) were obtained from ATCC and were cultured according to the manufacturer's instructions.

**Proliferation and senescence assays.** Proliferation assay in MEFs was performed by plating $10^4$ cells per well of 24-well plate in triplicate while that in human PCa cell lines was performed by plating $1-2 \times 10^4$ cells per well of 24-well plate in triplicate. Cells were treated with vehicle (dimethyl sulfoxide) control, AZD8186 (PI3K inhibitor, Astrazeneca) at 3 μM and 1 μM and GSI at aforementioned concentrations. Cell proliferation was monitored on days 0, 2, 4 and 6 whereby cells were fixed for 15 min in a solution of 10% buffered formalin washed with phosphate-buffered saline (pH7.2) and subsequently stained with 0.01% Crystal violet solution. Excessive staining was removed by washing with distilled water and drying the plates overnight. Crystal violet-stained cells were dissolved in 10% acetic acid solution for 30 min on a shaker and the extracted dye was read with a spectrophotometer at 590 nm. Cellular senescence in vitro was performed using the Senescence β-Galactosidase Staining Kit (Cell Signaling) as per the manufacturer's instructions.

**siRNA and shRNA transfection.** Human ADAM17-directed shRNA was obtained from Sigma. To prepare lentiviral particles, 293T human embryonic kidneys were transfected using Jetprime transfection reagents (Polyplus transfection) as per the manufacturer's instructions. PC3 cells were infected with the lentivirus from transfected 293T human embryonic kidneys and were subsequently selected using puromycin (2 µg ml$^{-1}$). shRNA: 5′-CCGGCCTATGTCGATGCTGAACAA ACTCGAGTTTGTTCAGCATCGACATAGGTTTTTG-3′ (Clone ID:NM_ 003183.3-2002s1c1).

Human CUX1a sequence: 5′-AACAGGAGGACACAAGGCAAAGCUG-3′ and CUX1b sequence: 5′-CAGGGUUUGUUUAAUACACUCCAUU-3′ were custom siRNA synthesized (Dharmacon). siRNA transfection was performed using Jetprime transfection reagents (Polyplus transfection) as per the manufacturer's instructions.

**WB and histology.** Human tissue microarrays were purchased from Biomax, Inc. (PR8011A,PR483B). The antibodies used for IHC analysis and WB were anti-activated Notch1 (Abcam) (IHC), Cleaved Notch1 (Val 1744) (Cell Signaling Technology; 1:250 dilution), Notch1 (D1E11) (Cell Signaling Technology; 1:1,000) (WB), PTEN (Cell Signaling Technology; 1:1,000) (WB), PTEN (51–2,400; Invitrogen) (IHC), Ki-67 (Clone SP6; Lab Vision) (IHC), HSP90 (Cell Signaling Technology; 1:1,000), ADAM10 (Abcam; 1:1,000), ADAM17 (Abcam; 1:1,000) (IHC/WB), CathepsinL (Abcam; 1:1,000) (WB), CUX1 a.a. 1,300 (Millipore, 1:2,500) and CUX1 a.a. 861 (Millipore; 1:1,000) (WB), Skp2 (Santa Cruz; 1:500) (WB), p27 (Santa Cruz; 1:500) (WB), Hes1 (Santa Cruz; 1:500) (WB), c-Myc (Santa Cruz; 1:500) (WB), p21 (Santa Cruz; 1:500) (WB) and β-actin (Sigma; 1:5,000) (WB). IF analysis in prostate tissues was performed by using Vimentin (Abcam, 1:350) and E-cadherin (BD Biosciences 1:400) antibodies.

**ChIP assay.** Cells were cultured up to a confluence of 90–95% and were crosslinked with 1% formalin for 10 min followed by addition of glycine for 5 min at room temperature. The culture medium was aspirated and the cells were washed twice with ice-cold phosphate-buffered saline. Nuclear extracts were sonicated using a Misonix 3,000 model sonicator to sheer crosslinked DNA to an average fragment size of ~500 bp. Sonicated chromatin was incubated for 16 h at 4 °C with γ-bind Plus sepharose beads (GE Healthcare) conjugated with either anti-CUX1 antibody (Santa Cruz; 200 µg per 0.1 ml) or IgG antibody (Millipore) by incubating overnight at 4 °C on a rotor. After incubation, beads were washed thoroughly and then centrifuged. The chromatin was eluted from the beads, and crosslinks were removed by incubation at 65 °C for 5 h. DNA was then purified using the QIAquick PCR Purification Kit (Qiagen). The ChIP primers for ADAM17 EpiTect ChIP quantitative PCR (qPCR) Primer Assay For Human ADAM17, NM_003183.4 ( − )02Kb (Qiagen) and PIKChIP1f sequence: 5′-GAGGAAGGAAGGTACTGAACC-3′ and PIKChIP1r sequence: 5′-CCTGTAACTAAGACATTTATCAGC-3′. ChIP qPCR was performed using KAPA SYBR FAST ABI qPCR Master Mix solution (KAPA Biosystem) on Step One Real-Time PCR systems (Applied Biosystems). Primers for ADAM17 used in the ChIP experiments were designed using the SABiosciences' proprietary database (DECODE, DECipherment Of DNA Elements).

**Quantitative real-time PCR.** Quantitative real-time PCR was performed on RNA extracted from cells and the respective tissues samples using Trizol (Invitrogen). Complementary DNA was prepared with SuperScript III First-Strand Synthesis SuperMix (Invitrogen). See Supplementary Table 3 for list of primers used for qRT-PCR. Quantitative real-time PCR was performed using KAPA SYBR FAST ABI qPCR Master Mix solution (KAPA Biosystem) on Step One Real-Time PCR systems (Applied Biosystems).

**Correlation analysis.** Correlation between gene-expression-derived values in the principle-component analysis PCa data sets was carried out using Pearson's correlation test, which estimates a correlation value 'r' and a significance P value ($r > 0 < 1$, direct correlation; $r < 0 > 1$, inverse correlation). Correlation was also performed in tissue microarray staining evaluation using the estimated percentage of positively stained cells. Pearson's r from correlation analyses was calculated to assess potential positive ($r > 0$) or negative ($r < 0$) linear correlations between the gene expression levels of PTEN and ADAM17 and between PTEN and HES1 in the primary tumour biopsies ($n = 49$) comprised in the Grasso human PCA data set (GSE35988). Two tailed P values $< 0.05$ were considered significant.

**Gene expression profiling.** Gene expression profiling was carried out using the MouseRef-8 v2.0 Expression BeadChip (Illumina, San Diego, CA, USA), following the manufacturer's protocol. Arrays were read on an Illumina HiScanSQ system. Data were first extracted with the Illumina GenomeStudio software and then imported in Genomics Suite 6.4 (Partek Incorporated, Saint Louis, MO, USA) and quantile normalized. Transcripts with differences in expression were identified by analysis of variance. Enrichment analysis was performed using GSEA[64]. Raw data have been deposited in National Center for Biotechnology Information's Gene Expression Omnibus (GEO) and are accessible through GEO accession (GSE76822). GSEA was performed on entire gene list ranked according to fold changes observed between $Pten^{pc+/+}$ and $Pten^{pc-/-}$ mice and also between $Pten^{pc-/-}$ and $Pten^{pc-/-}$ treated with PF-03084014 mice. Functional analysis

was performed on the collapsed gene symbol list using GSEA with the MSigDB-v4.0 (Molecular Signatures Database)[65] C2-C7 gene sets. Gene sets with false discovery rate $< 0.25$ and Normalized Enriched Score[66] $> 1.25$ or $< -1.25$ were considered significantly enriched. The VILIMAS_Notch_Targets[67] gene set include 52 genes upregulated in bone marrow progenitors by constitutively active NOTCH1. Pearson's correlation was used to study the association among genes in terms of gene expression. Analyses were performed using the R environment (R Studio console; RStudio, Boston, MA, USA). A P value $< 0.05$ was considered statistical significant.

**Secrete-pair dual luminescence assay.** Cells were co-transfected with ADAM17 promoter reporter clone (HPRM15027, Genecopoeia) along with control vector (pXJ Vector) and p110 CUX1 (pXJ p110). Media from these transfected cells from at different specified time points were collected after changing the media post-transfection. The secreted luciferase (GLuc and SEAP) was measured and analysed as per the manufacturer's guide (SPDA-D010, Genecopoeia).

**IF analysis.** IF images were acquired on a Leica TCS SP5 confocal microscope, using a ×40/1.25N.A. objective (Leica HCX PL APO lambda blue ×40/1.25 oil UV). Several fields of views were acquired with tiling scan function in order to get an area of $1,000 \times 1,000$ µm$^2$. Image analysis was performed measuring total fluorescence of TF both in nuclear and cytoplasmic region, with a customized pipeline in the CellProfiler software[66]. Images for tissue samples stained for Vimentin/E-cadherin were acquired on a Leica TCS SP5 confocal microscope using ×10/1.25 oil.

**Statistical analysis.** For each independent *in vitro* experiment, at least three technical replicates were performed with an exception in WB analysis. In the *in vitro* experiments, data groups were assessed for normal distribution and Student's *t*-test was performed for paired comparison. The *n* values represent the number of mice used for the study of genetically engineered mouse model analysis and preclinical trials using PF-03084014.

Data analysis was performed using a two-tailed unpaired Student's *t*-test. Values are expressed as mean ± s.e.m. (*$P < 0.05$; **$P < 0.01$; ***$P < 0.001$).

**Data availability.** The data that support the findings of this study are available from the corresponding author upon reasonable request. Full scans of WBs are available in Supplementary Fig. 9.

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

# ARTICLE

15. Weng, A. P. et al. Activating mutations of NOTCH1 in human T cell acute lymphoblastic leukemia. Science 306, 269–271 (2004).

16. Maraver, A. et al. Therapeutic effect of gamma-secretase inhibition in KrasG12V-driven non-small cell lung carcinoma by derepression of DUSP1 and inhibition of ERK. Cancer Cell 22, 222–234 (2012).

17. Radtke, F. & Raj, K. The role of Notch in tumorigenesis: Oncogene or tumour suppressor? Nat. Rev. Cancer 3, 756–767 (2003).

18. Roy, M., Pear, W. S. & Aster, J. C. The multifaceted role of Notch in cancer. Curr. Opin. Genet. Dev. 17, 52–59 (2007).

19. Sulis, M. L., Saftig, P. & Ferrando, A. A. Redundancy and specificity of the metalloprotease system mediating oncogenic NOTCH1 activation in T-ALL. Leukemia 25, 1564–1569 (2011).

20. Brou, C. et al. A novel proteolytic cleavage involved in Notch signaling: the role of the disintegrin-metalloprotease TACE. Mol. Cell 5, 207–216 (2000).

21. Mumm, J. S. et al. A ligand-induced extracellular cleavage regulates gamma-secretase-like proteolytic activation of Notch1. Mol. Cell 5, 197–206 (2000).

22. Kopan, R. & Ilagan, M. X. The canonical Notch signaling pathway: unfolding the activation mechanism. Cell 137, 216–233 (2009).

23. Murphy, G. The ADAMs: signalling scissors in the tumour microenvironment. Nat. Rev. Cancer 8, 929–941 (2008).

24. Ding, Z. et al. SMAD4-dependent barrier constrains prostate cancer growth and metastatic progression. Nature 470, 269–273 (2011).

25. Li, J. et al. PTEN, a putative protein tyrosine phosphatase gene mutated in human brain, breast, and prostate cancer. Science 275, 1943–1947 (1997).

26. Taylor, B. S. et al. Integrative genomic profiling of human prostate cancer. Cancer Cell 18, 11–22 (2010).

27. Steck, P. A. et al. Identification of a candidate tumour suppressor gene, MMAC1, at chromosome 10q23.3 that is mutated in multiple advanced cancers. Nat. Genet. 15, 356–362 (1997).

28. Shen, M. M. & Abate-Shen, C. Molecular genetics of prostate cancer: new prospects for old challenges. Genes Dev. 24, 1967–2000 (2010).

29. Alimonti, A. et al. Subtle variations in Pten dose determine cancer susceptibility. Nat. Genet. 42, 454–U136 (2010).

30. Trotman, L. C. et al. Pten dose dictates cancer progression in the prostate. PLoS Biol. 1, 385–396 (2003).

31. Palomero, T. et al. Mutational loss of PTEN induces resistance to NOTCH1 inhibition in T-cell leukemia. Nat. Med. 13, 1203–1210 (2007).

32. Goulet, B. et al. A cathepsin L isoform that is devoid of a signal peptide localizes to the nucleus in S phase and processes the CDP/Cux transcription factor. Mol. Cell 14, 207–219 (2004).

33. Cadieux, C. et al. Mouse mammary tumor virus p75 and p110 CUX1 transgenic mice develop mammary tumors of various histologic types. Cancer Res. 69, 7188–7197 (2009).

34. Moon, N. S., Berube, G. & Nepveu, A. CCAAT displacement activity involves CUT repeats 1 and 2, not the CUT homeodomain. J. Biol. Chem. 275, 31325–31334 (2000).

35. Ramdzan, Z. M. & Nepveu, A. CUX1, a haploinsufficient tumour suppressor gene overexpressed in advanced cancers. Nat. Rev. Cancer 14, 673–682 (2014).

36. Chen, Z. B. et al. Crucial role of p53-dependent cellular senescence in suppression of Pten-deficient tumorigenesis. Nature 436, 725–730 (2005).

37. Ranganathan, P., Weaver, K. L. & Capobianco, A. J. Notch signalling in solid tumours: a little bit of everything but not all the time. Nat. Rev. Cancer 11, 338–351 (2011).

38. Olsauskas-Kuprys, R., Zlobin, A. & Osipo, C. Gamma secretase inhibitors of Notch signaling. Onco Targets Ther. 6, 943–955 (2013).

39. Papayannidis, C. et al. A Phase 1 study of the novel gamma-secretase inhibitor PF-03084014 in patients with T-cell acute lymphoblastic leukemia and T-cell lymphoblastic lymphoma. Blood Cancer J. 5, e350 (2015).

40. Schouwey, K. & Beermann, F. The Notch pathway: hair graying and pigment cell homeostasis. Histol. Histopathol. 23, 609–619 (2008).

41. Moriyama, M. et al. Notch signaling via Hes1 transcription factor maintains survival of melanoblasts and melanocyte stem cells. J. Cell Biol. 173, 333–339 (2006).

42. Lin, H. K. et al. Skp2 targeting suppresses tumorigenesis by Arf-p53-independent cellular senescence. Nature 464, 374–379 (2010).

43. Yashiro-Ohtani, Y. et al. Pre-TCR signaling inactivates Notch1 transcription by antagonizing E2A. Genes Dev. 23, 1665–1676 (2009).

44. Weng, A. P. et al. c-Myc is an important direct target of Notch1 in T-cell acute lymphoblastic leukemia/lymphoma. Genes Dev. 20, 2096–2109 (2006).

45. Girard, L. et al. Frequent provirus insertional mutagenesis of Notch1 in thymomas of MMTVD/myc transgenic mice suggests a collaboration of c-myc and Notch1 for oncogenesis. Genes Dev. 10, 1930–1944 (1996).

46. Robey, E. et al. An activated form of Notch influences the choice between CD4 and CD8 T cell lineages. Cell 87, 483–492 (1996).

47. Cui, D. et al. Notch pathway inhibition using PF-03084014, a gamma-secretase inhibitor (GSI), enhances the antitumor effect of docetaxel in prostate cancer. Clin. Cancer Res. 21, 4619–4629 (2015).

48. Bray, S. J. Notch signalling: a simple pathway becomes complex. Nat. Rev. Mol. Cell Biol. 7, 678–689 (2006).

49. Bozkulak, E. C. & Weinmaster, G. Selective use of ADAM10 and ADAM17 in activation of Notch1 signaling. Mol. Cell. Biol. 29, 5679–5695 (2009).

50. Vadnais, C. et al. Long-range transcriptional regulation by the p110 CUX1 homeodomain protein on the ENCODE array. BMC Genomics 14, 258 (2013).

51. Toso, A. et al. Enhancing chemotherapy efficacy in Pten-deficient prostate tumors by activating the senescence-associated antitumor immunity. Cell Rep. 9, 75–89 (2014).

52. Skaar, J. R., Pagan, J. K. & Pagano, M. SCF ubiquitin ligase-targeted therapies. Nat. Rev. Drug Discov. 13, 889–903 (2014).

53. Palumbo, A. et al. Bortezomib, doxorubicin and dexamethasone in advanced multiple myeloma. Ann. Oncol. 19, 1160–1165 (2008).

54. Adams, J. The proteasome: a suitable antineoplastic target. Nat. Rev. Cancer 4, 349–360 (2004).

55. Wang, Z. et al. Skp2: a novel potential therapeutic target for prostate cancer. Biochim. Biophys. Acta 1825, 11–17 (2012).

56. Szalad, A., Katakowski, M., Zheng, X., Jiang, F. & Chopp, M. Transcription factor Sp1 induces ADAM17 and contributes to tumor cell invasiveness under hypoxia. J. Exp. Clin. Cancer Res. 28, 129 (2009).

57. Charbonneau, M. et al. Hypoxia-inducible factor mediates hypoxic and tumor necrosis factor alpha-induced increases in tumor necrosis factor-alpha converting enzyme/ADAM17 expression by synovial cells. J. Biol. Chem. 282, 33714–33724 (2007).

58. Li, R. et al. High glucose up-regulates ADAM17 through HIF-1alpha in mesangial cells. J. Biol. Chem. 290, 21603–21614 (2015).

59. Mailly, F. et al. The human cut homeodomain protein can repress gene expression by two distinct mechanisms: active repression and competition for binding site occupancy. Mol. Cell. Biol. 16, 5346–5357 (1996).

60. Wong, C. C. et al. Inactivating CUX1 mutations promote tumorigenesis. Nat. Genet. 46, 33–38 (2014).

61. Cacquevel, M. et al. Rapid purification of active gamma-secretase, an intramembrane protease implicated in Alzheimer's disease. J. Neurochem. 104, 210–220 (2008).

62. Wu, F. et al. Novel gamma-secretase inhibitors uncover a common nucleotide-binding site in JAK3, SIRT2, and PS1. FASEB J. 24, 2464–2474 (2010).

63. Alattia, J. R. et al. Mercury is a direct and potent gamma-secretase inhibitor affecting Notch processing and development in Drosophila. FASEB J. 25, 2287–2295 (2011).

64. Thompson, M. A. et al. Differential gene expression in anaplastic lymphoma kinase-positive and anaplastic lymphoma kinase-negative anaplastic large cell lymphomas. Hum. Pathol. 36, 494–504 (2005).

65. Liberzon, A. et al. Molecular signatures database (MSigDB) 3.0. Bioinformatics 27, 1739–1740 (2011).

66. Carpenter, A. E. et al. CellProfiler: image analysis software for identifying and quantifying cell phenotypes. Genome Biol. 7, R100 (2006).

67. Vilimas, T. et al. Targeting the NF-kappaB signaling pathway in Notch1-induced T-cell leukemia. Nat. Med. 13, 70–77 (2007).

## Acknowledgements

We thank Pfizer for providing us with the γ-secretase inhibitor PF-03084014 and the financial support. This work was supported by, European Research Council (ERC) starting grant (261342) and ERC consolidator (683136), Josef Steiner Foundation, Helmut Horten Foundation, Pfizer (WI173087) and Krebsliga (KFS 3505-08-2014) grants. We are grateful for helpful comments and various discussions with Bertoni and Fraering groups and all the members of the Alimonti group. We also thank Alain Nepveu for providing the p110 CUX1 overexpression plasmid and protein lysates and slides from p110 CUX1 mammary mouse model and the IOR and IRB staff members of the core animal facility.

## Author contributions

A. Alimonti, A. Revandkar and A.T. conceived the project and wrote the manuscript. A.Alimonti, A. Revandkar and A.T. designed and analysed data. A. Revandkar, M.L.P., E.P., A. Alajati and A.T. performed experiments. H.G., M.D. and P.F. designed, performed and analysed data for the γ-secretase assay. N.D. performed bioinformatic correlation analysis on PCa data set. A. Rinaldi and F.B. designed, performed and analysed gene expression experiments in Pten tumours. A. Arribas performed bioinformatic correlation analysis in SMZL data set. R.D. acquired, analysed and quantified the immunofluorescence images. S.P. and M.L. performed IHC experiments. H.G. and P.F. provided additional reagents and analysis tools for the γ-secretase assay.

## Additional information

**Accession number:** The Gene Set Enrichment Analysis data has been deposited in NCBI's Gene Expression Omnibus under GEO: GSE76822.

**Competing financial interests:** The authors declare no competing financial interests.

