## [Peer Review File · Nature Communications]

Reviewers' comments:

Reviewer #1 (Notch/PTEN expert)(Remarks to the Author):

The manuscript demonstrates convincingly that loss of PTEN in prostate cancer models results in increased Notch1 cleavage and activation through CUX1-mediated transactivation of ADAM17, followed by ligand-independent Notch activation. In general, the findings are innovative and data are of good quality. Most of the conclusions are supported by experimental evidence, with one exception (see below). The main strengths of the manuscript are the novel observation that CUX1 mediates increased ADAM17 expression, and that ADAM17 is required for Notch1 activation in these models. The animal model experiments are elegant. Additionally, these observations suggest a novel potential therapeutic target in this Notch activation pathway (Cathepsin L). The manuscript could be further strengthened by addressing the following issues:

- In the model proposed by the authors and shown in Supplemental Figure 6, PTEN loss of function (LOF) results in increased expression of Cathepsin L, which in turn cleaves CUX1 p200 producing CUX1 p100. This in turn results in increased expression of ADAM17 and (presumably) ligand-independent Notch1 activation. In this model, the effect of PTEN LOF is independent of its well-known consequences on the PI3K-AKT pathway. There is no evidence in the manuscript that this effect is indeed PI3K- or AKT-independent. This is an important point that needs to be clarified by additional experimental data.
- Figure 1C: it appears that the total amount of Notch1, not just NICD, is increased in PTEN -/- prostates. Is this due to transcriptional upregulation of NOTCH1? The authors should discuss possible mechanisms for this. In some systems, Notch1 induces its own expression, and that may explain this observation.
- Figure 1H: Western blots are not labeled. Although the legend explains what was analyzed, the figure should be labeled.
- Figure 4D: PTEN-positive cells DU-145 appear to have very high levels of ADAM17, comparable to those of PTEN-null cell lines. This suggests that PTEN is only one of the possible mechanisms that upregulate ADAM17 in PC cells.

Reviewer #2 (prostate cancer expert) (Remarks to the Author):

Summary:

Notch signaling has been associated with prostate cancer progression and therapeutic resistance. This study suggests that loss of PTEN loss up-regulates expression of the metalloprotease ADAM17, resulting in the cleavage of Notch1 receptor and activation of Notch signaling. Using conditional mouse models to inactivate Pten and Notch1 in the rodent prostate, the study proposes that Pten-deficient prostate tumors are addicted to Notch signaling. Furthermore, pharmacological inhibition of γ -secretase promotes cellular senescence and growth arrest in Pten-null and Pten/Trp53-null prostate tumors. Therefore it is concluded that γ -secretase inhibitors could be a viable treatment of patients with advanced prostate cancer.

Comments:

1. The Introduction Section contains a brief description of Notch and Pten. The concept that Pten modulates Notch, Notch target gene expression, and tumor formation would be strengthened providing more specific information on the major genes/proteins to be investigated in this manuscript. This would include ADAM17 (which is not even named in the Introduction) and Cux1 (which suddenly appears without any background information as to what this protein is), and any known links between these proteins and PTEN and/or Notch signaling.
2. Please define all acronyms (AICD, etc).

3. Reference 23 is quoted as a basis for the rationale that, "In an attempt to identify the regulators of NOTCH signaling in PCa, we have found that several NOTCH target genes are strongly up-regulated in prostate tumors characterized by loss of the tumor suppressor gene PTEN.....(page 3)". Reference 23 refers to a study in which Pten/Smad interactions were investigated. Therefore the connection to Notch signaling is unclear.

4. The methodology is exceedingly brief. There should be sufficient information and or references to allow another lab to duplicate an experiment. Examples which would benefit from additional information would include Gene Set Enrichment Analysis (GSEA) and VILIMAS_Notch_Targets_UP (Figures 1d and 2h).

5. The experimental design and the description of the results using the PTEN and Notch knockdown mouse models are difficult to evaluate for the following observations:

(i) The *Pten^{pc}/-* model. Prostate tumor formation is highly dependent on the levels by which *Pten* expression is knocked down. Therefore, it would be reasonable to determine the penetrance of knocked-down expression and how long it would take for the male mice to develop cancerous lesions, if any, in the prostate. Previously, Alimonti et al (Nature Genetics, 2010) published that while decreased PTEN expression was strongly associated with human prostate cancer progression, *Pten^{hy/+}* male mice did not develop prostate intraepithelial neoplasia (PIN). Therefore, it is not clear why specific time points for histopathological analysis (for example 12 weeks in Figure 1) were selected. Knocking down *Pten* expression alone appears to be, at best, a model of prostate cell hyperplasia, i.e., increased epithelial cell proliferation. In addition, the degree of hyperplasia would be dependent on the penetrance by which the *Pten* gene was floxed out. This could account for the variable lobe sizes observed in Figure 2b. Strain differences could also make a difference in lobe size and gene expression levels. Were the wt and floxed models all on the same genetic background?

(ii) The study did not rigorously evaluate the histopathology of prostate cancer development and progression in *Notch1^{pc}/-* and *Pten^{pc}/-;Notch1^{pc}/-* mice as compared to *Pten^{pc}/-* and wt controls. Therefore the etiology of prostate cancer development in these models is not clear.

(iii) Given that tissues derived from Cre-lox models exhibit varying degrees of knockdown gene expression, the statistical power of this study appears low with numbers of animals per group ranging from n=3 to n=5.

6. Figure 3 uses the *Pten^{pc}/-; Trp53^{pc}/-* mouse model to determine the efficacy of the Notch inhibitor PF-03084014. This model is appropriate since *Pten^{pc}/-; Trp53^{pc}/-* male mice are reported to develop bona fide adenocarcinoma. Unfortunately it is difficult to evaluate the efficacy of treatment since the experiment did not contain wt, *Pten^{pc}/-* and/or *Trp53^{pc}/-* control groups which would demonstrate that PF-03084014 treatment did not affect normal prostate morphology or weights. Moreover the length of treatment and rationale for the dose used was not given. It would have been more informative to perform a dose-response curve or a time course to demonstrate the efficacy of treatment as defined by decrease tumor weight, decreased cell proliferation, etc.

In addition, the criteria for determining cell invasion were not defined. Since prostatic glandular structure is highly convoluted, how was it determined that a metastatic acinar structure had arisen, separated from the parent gland, and had now invaded into the prostatic stroma?

Were any distant metastasis observed and if yes, did their number also decrease with PF-03084014 treatment?

7. Figure 4, page 8/9, the statement, "ADAM17 was specifically up-regulated in PCa cell lines

having reduced levels of PTEN". This observation appears overstated since DU-145 cells, which express similar levels of Pten as 22Rv1 cells, also expressed similar levels of Adam17 as Pten-negative LNCaP and PC3 cells (Figure 4d-f).

And the next statement, "This was also associated to increased level of NICD1...." Again, this observation appears to be an overstatement since DU-145 cells which express Pten also express similar levels of NICD1 as Pten-negative PC3 cells.

8. Figure 4g,h, page 9, the statement, "Our analysis revealed that the majority of samples displaying low levels of PTEN stained positive for ADAM17". This analysis appears very basic. Since these are tissue microarrays, they would likely come with data regarding tumor grade, Gleason score, etc. Therefore, it would have been more informative to demonstrate that loss of Pten and increased in Adam17 expression directly correlated with increased tumor grade/Gleason score and tumor progression.

9. Figure 5. The data are very interesting and present a nice analysis of the relationship between Adam17 and Cux1 expression.

10. Figure 6. This figure is confusing since it presents data in breast cancer, not prostate cancer. The Ptenpc^{-/-}; Trp53pc^{-/-} mouse model might have been more useful, especially for panels d-f.

11. A general source of difficulty in evaluating this study is the use of numerous prostate cell lines without providing any rationale as to why one cell line is used over another. Moreover, it is not clear why the data was never confirmed using a second cell line which expressed similar levels of the protein of interest.

We thank the Referees for their constructive comments, which have helped to further improve the quality of our manuscript. We have now reorganized our manuscript on the basis of a number of critical new findings. Furthermore, we have addressed in full the previous critiques and constructive suggestions of the referees, which have helped tremendously at strengthening and corroborating our original conclusions, specifically, the pre-clinical relevance of our findings.

Taken together our findings lead to a number of relevant conclusions:

- 1) PTEN deficiency leads to the up-regulation of ADAM17, which in turn activates NOTCH signaling.
- 2) In PTEN null tumors, up-regulation of ADAM17 is mediated by an oncogenic isoform of the transcription factor CUX1 (Cut-Like Homeobox 1) (p110 CUX1), which binds to the promoter of ADAM17 favoring its transcription. Therefore CUX1 is a novel regulator of the NOTCH pathway.
- 3) Finally, we have run a series of pre-clinical trials in mice using different prostate cancer mouse models (*Pten*^{pc/-} and *Pten*^{pc/-}; *Trp53*^{pc/-}). In these experiments we have found that advanced PTEN-deficient tumors are addicted to the NOTCH signaling.

Thus, our study provides a novel rationale for targeting NOTCH signaling in prostate cancer.

Referee 1:

We thank the Referee for his/her constructive comments, which have helped to further improve the quality of our manuscript. In this rebuttal, and in our revised manuscript, we have incorporated all the suggestions of this reviewer to strengthen the relevance of our previous findings. As suggested by this referee, we have performed key experiments to better understand the mechanism of activation and upregulation of Notch signalling in PTEN deficient tumours and cancer cell lines.

Major points:

- 1. In the model proposed by the authors and shown in Supplemental Figure 6, PTEN loss of function (LOF) results in increased expression of Cathepsin L, which in turn cleaves CUX1 p200 producing CUX1 p100. This in turn results in increased expression of ADAM17 and (presumably) ligand-independent Notch1 activation. In this model, the effect of PTEN LOF is independent of its**

well-known consequences on the PI3K-AKT pathway. There is no evidence in the manuscript that this effect is indeed PI3K- or AKT-independent. This is an important point that needs to be clarified by additional experimental data.

To address this point we have analyzed four prostate cancer cell lines namely 22Rv1, DU145, LNCaP and PC3 and treated them with AZD8186, a potent PI3K inhibitor¹. As shown in the paper cell lines harboring both heterozygous (DU-145) and homozygous loss of function of PTEN (LNCaP and PC3) have elevated levels of ADAM17 and NICD1 (Fig. 4d). Inhibition of the PI3K-AKT pathway in these cells decreased phosphorylation of AKT without reducing NOTCH1-intracellular domain (NICD1) and ADAM17 protein

Rebuttal Figure 1: Activated Notch1 upregulates its own gene expression

levels (see new Supplementary Fig. 5). These data suggest that the activation of NOTCH signaling pathway upon loss of PTEN is PI3K/AKT-independent as suggested by this Referee.

2. Figure 1C: it appears that the total amount of Notch1, not just NICD, is increased in PTEN -/- prostates. Is this due to transcriptional upregulation of NOTCH1? The authors should discuss possible mechanisms for this. In some systems, Notch1 induces its own expression, and that may explain this observation

We thank the referee for this keen observation about possible transcriptional up-regulation of Notch1 in this context. We agree with the referee that activation of the NOTCH pathway may result in the upregulation of NOTCH1 levels as previously reported in other papers²⁻⁵. As pointed out by the referee, this explains the data of Fig. 1b and c. The RT-PCR data obtained from *Pten*^{PC+/+} prostate tissues and *Pten*^{PC-/-} tumours demonstrate that the Notch1 upregulation is transcriptionally mediated. Interestingly, treatment of *Pten*^{PC-/-} tumors with the γ -secretase inhibitor PF-03084104, as shown by downregulation of *Hes1*, blocked the expression of Notch1 validating this hypothesis (Fig.

1 of this rebuttal). This data, in line with the referees' speculation, suggests that Notch1 activation induces its own expression. As suggested by the referee, in the new version of the manuscript (see text in yellow in discussion) we have considered this possible mechanism. If this referee thinks that this figure should be included in the final version of the manuscript we will be happy to include it.

3. Figure 1H: Western blots are not labeled. Although the legend explains what was analyzed, the figure should be labeled.

We apologize to the referee for missing out the labelling of the western. In the revised version of manuscript we have corrected our mistake.

4. Figure 4D: PTEN-positive cells DU-145 appear to have very high levels of ADAM17, comparable to those of PTEN-null cell lines. This suggests that PTEN is only one of the possible mechanisms that upregulate ADAM17 in PC cells.

We thank the referee for this question and we apologize for not being clear regarding this point. We would like to shed light on the fact that DU-145 cell line carries a loss of function mutation in one of the PTEN alleles. As previously documented, this leads to partial loss of function of PTEN⁶. Therefore, although the total protein levels of PTEN in 22Rv1 and DU145 are similar, DU145 have only one functional allele of PTEN. In the new version of the manuscript we have clarified this point.

Referee 2:

We thank the Referee for his/her constructive comments, which have helped us to further improve the quality of our manuscript. In this rebuttal, and in our revised manuscript, we have incorporated the suggestions of this reviewer to strengthen the relevance of our findings. As suggested by this Referee we have included all the controls for the mouse model and preclinical trials along with additional clinical data analysis

1. The Introduction Section contains a brief description of Notch and Pten. The concept that Pten modulates Notch, Notch target gene expression, and tumor formation would be strengthen providing more specific information on the major genes/proteins to be investigated in this manuscript. This would include ADAM17 (which is not even named in the Introduction) and Cux1

(which suddenly appears without any background information as to what this protein is), and any known links between these proteins and PTEN and/or Notch signaling.

We thank the referee for this comment and we apologize for the brief description of ADAM17 and CUX1 in the introduction of the paper. In the new version of the manuscript, we have now included a detailed introduction on ADAM17 and CUX1.

2. Please define all acronyms (AICD, etc).

We apologize for this mistake. The meaning of the acronyms AICD and NICD are now included in the revised version of our manuscript.

3. Reference 23 is quoted as a basis for the rationale that, "In an attempt to identify the regulators of NOTCH signaling in PCa, we have found that several NOTCH target genes are strongly up-regulated in prostate tumors characterized by loss of the tumor suppressor gene PTEN.....(page 3)". Reference 23 refers to a study in which Pten/Smad interactions were investigated. Therefore the connection to Notch signaling is unclear.

We apologize to the referees for not being clear regarding this point. We have now modified the text explaining the reason why we have quoted the manuscript of the reference 23.

4. The methodology is exceedingly brief. There should be sufficient information and or references to allow another lab to duplicate an experiment. Examples which would benefit from additional information would include Gene Set Enrichment Analysis (GSEA) and VILIMAS_Notch_Targets_UP (Figures 1d and 2h).

We thank the referee for his comment. In the new version of the manuscript we have included a more detailed explanation of the methods, including a better description of the GSEA (see text in yellow).

5. The experimental design and the description of the results using the PTEN and Notch knockdown mouse models are difficult to evaluate for the following observations:

(1) The *Pten*^{pc/-} model. Prostate tumor formation is highly dependent on the levels by which *Pten* expression is knocked down. Therefore, it would be reasonable to determine the penetrance of knocked-down expression and how long it would take for the male mice to develop cancerous lesions, if any, in the prostate. Previously, Alimonti et al (Nature Genetics, 2010) published that while decreased PTEN expression was strongly associated with human prostate cancer progression, *Pten* ^{hy/+} male mice did not develop prostate intraepithelial neoplasia (PIN). Therefore, it is not clear why specific time points for histopathological analysis (for example 12 weeks in Figure 1) were selected. Knocking down *Pten* expression alone appears to be, at best, a model of prostate cell hyperplasia, i.e., increased epithelial cell proliferation. In addition, the degree of hyperplasia would be dependent on the penetrance by which the *Pten* gene was floxed out. This could account for the variable lobe sizes observed in Figure 2b. Strain differences could also make a difference in lobe size and gene expression levels. Were the wt and floxed models all on the same genetic background?

We thank the referee for this comment and we apologize for not being clear regarding these points. Probably the referee is not familiar with this model and we did not sufficiently describe it in the manuscript. As previously published from our group, and from the group of PP Pandolfi⁷⁻⁹, while *Pten* ^{hy/+} male (80% *Pten* expression) do not develop prostate lesions, both *Pten* ^{+/-} mice (50% *Pten* expression) and *Pten* null mice (0% *Pten* expression) develop prostate tumor lesions. In particular the *Pten* null prostate conditional (*Pten*^{pc/-}) mice used in this study develop 100% penetrant high-grade prostatic intraepithelial neoplasia (PIN) at 8 weeks of age. These high-grade pre-tumoral lesions rapidly progress to invasive prostate cancer in mice from 15 weeks of age, fully recapitulating the natural history of human prostate cancer⁹. In Fig. 1f of the manuscript we have selected specific time points to analyze different stages of tumor progression in these mice. As reported in this figure, deletion of *Notch1* in *Pten* null mice promotes a strong tumor inhibition. Indeed, while *Pten*^{pc/-} mice from 16 weeks of age develop invasive prostate cancer, the *Pten*^{pc/-}; *Notch1*^{pc/-} mice did not develop invasive prostate tumors. Moreover, as reported in previous papers, in the *Pten*^{pc/-} model,

inactivation of *Pten* occurs in all the cases analyzed and in all the mouse prostatic lobes, anterior (AP), dorsolateral (DLP) and ventral prostates⁹. Indeed, as also represented in the new Fig. 1a of the manuscript, and reported in several previous paper⁷⁻⁹; immunohistochemistry staining for Pten in *Pten*^{pc/-} mice at 8-10 weeks of age shows that Pten is completely lost in these tumors. Finally, we used the same genetic background (BL6/C57) for all the mouse models used in this study.

(2) The study did not rigorously evaluate the histopathology of prostate cancer development and progression in *Notch1pc/-* and *Ptenpc/-*; *Notch1pc/-* mice as compared to *Ptenpc/-* and wt controls. Therefore the etiology of prostate cancer development in these models is not clear.

We thank the referee for this suggestion and we apologize for this omission. In the revised version of our manuscript (Fig1e-g) we have included the histopathological analysis of both *Notch1*^{pc/-} and *WT* prostate tissues along with *Pten*^{pc/-} and *Pten*^{pc/-}; *Notch1*^{pc/-} tumors. Moreover, we have included a new quantification of this analysis along with the Ki67 staining in this figure. Of note, we did not observe any changes in the histology and Ki67 staining in both the WT and *Notch1*^{pc/-} prostate tissues analyzed. Finally, we have modified the text of the manuscript to include the description of these data (see text in yellow in the new version of the manuscript).

(3) Given that tissues derived from Cre-lox models exhibit varying degrees of knockdown gene expression, the statistical power of this study appears low with numbers of animals per group ranging from n=3 to n=5.

As reported above (see point 1 referee2), the percentage of Pten recombination in these mice is 100%. Moreover, following the referee suggestions, we have now added additional mice to the study n=5 for each group. Given the number of time points analyzed, we believed that this analysis has been performed with significant statistical power.

6. Figure 3 uses the *Ptenpc/-*; *Trp53pc/-* mouse model to determine the efficacy of the Notch inhibitor PF-03084014. This model is appropriate since *Ptenpc/-*; *Trp53pc/-* male mice are reported to develop bona fide adenocarcinoma. Unfortunately it is difficult to evaluate the efficacy of treatment since the experiment did not contain wt, *Ptenpc/-* and/or *Trp53pc/-* control

groups which would demonstrate that PF-03084014 treatment did not affect normal prostate morphology or weights. Moreover the length of treatment and rationale for the dose used was not given. It would have been more informative to perform a dose-response curve or a time course to demonstrate the efficacy of treatment as defined by decrease tumor weight, decreased cell proliferation, etc.

We thank the referee for this suggestion. In the revised version of the figures we have now included these controls (see new Figure 3 and Supplementary Fig. 3a and b). As reported in the new version of the manuscript, we didn't observe any difference in the histology and proliferative index of *WT* and *Trp53^{pc/-}* prostates in mice treated with control and PF-03084014. In the new version of the manuscript, we have also included a description of the length of treatment and the dose used (see methods section). Treatment with PF-03084014 was started at weeks 15 and last for 5 weeks. The decision to use this dose and scheme of administration was based on toxicological data obtained in previous pre-clinical trials¹⁰⁻¹³ and the recommendations of Pfizer that has supplied us with the γ -secretase inhibitor PF-03084014.

We would like to highlight that the preclinical trial of γ -secretase inhibitor PF-03084014 was mainly performed to determine the anti-tumor efficacy in treating PTEN-deficient prostate cancer patients and it was not our intention to propose this treatment as a preventive agent in healthy subjects. However, we believe that this data, which emphasizes on the selectivity of PF-03084014 in targeting Pten-deficient cells without affecting normal prostate, could be of same potential interest for the field.

In addition, the criteria for determining cell invasion were not defined. Since prostatic glandular structure is highly convoluted, how was it determined that a metastatic acinar structure had arisen, separated from the parent gland, and had now invaded into the prostatic stroma?

We apologize to the referee for not being precise about this point. In the new version of the manuscript we have included Immunofluorescence (IF) analysis for Vimentin (stromal marker) and E-Cadherin (epithelial marker) staining in *Pten^{pc/-}*; *Trp53^{pc/-}* tumors and control prostates untreated and treated with PF-03084014 (see new Fig. 3a). As the referee will appreciate, in untreated *Pten^{pc/-}*; *Trp53^{pc/-}* tumors we detect areas of epithelial cell invasion in the stroma of the tumors, high tumor grade and abundant reactive stroma (see Fig.3a). Strikingly treatment with PF-03084014 reduced

both the tumor invasiveness and reactive stroma in both *Pten*^{pc-/-} and *Pten*^{pc-/-}; *Trp53*^{pc-/-} tumors (Fig. 3a and Supplementary Fig. 3b). The text of the manuscript has been modified to include the description of these new figures (see text in yellow).

Were any distant metastasis observed and if yes, did their number also decrease with PF-03084014 treatment?

We thank the referee for this question. As previously reported the *Pten*^{pc-/-}; *Trp53*^{pc-/-} tumors do not metastasize even at late stage⁹.

7. Figure 4, page 8/9, the statement, "ADAM17 was specifically up-regulated in PCa cell lines having reduced levels of PTEN". This observation appears overstated since DU-145 cells, which express similar levels of Pten as 22Rv1 cells, also expressed similar levels of Adam17 as Pten-negative LNCaP and PC3 cells (Figure 4d-f). And the next statement, "This was also associated to increased level of NICD1...." Again, this observation appears to be an overstatement since DU-145 cells, which express Pten, also express similar levels of NICD1 as Pten-negative PC3 cells.

We apologize to the referee for this mistake. As addressed above (see point 4, referee 1), DU-145 cell line possesses one functional allele of PTEN. In the paper, we used this cell line to show that even partial loss of function of PTEN can trigger Notch pathway activation. In the new version of the manuscript we have change this mistake (see text in yellow), explaining that even if 22Rv1 and DU-145 have similar levels of PTEN, DU-145 have only one functional allele of PTEN.

8. Figure 4g,h, page 9, the statement, "Our analysis revealed that the majority of samples displaying low levels of PTEN stained positive for ADAM17". This analysis appears very basic. Since these are tissue microarrays, they would likely come with data regarding tumor grade, Gleason score, etc. Therefore, it would have been more informative to demonstrate that loss of Pten and increased in Adam17 expression directly correlated with increased tumor grade/Gleason score and tumor progression.

We thank the referee for this constructive criticism about the correlation of PTEN and ADAM17 levels with tumor grade and Gleason score. We have now included these data in Fig. 4i of the new version of the manuscript.

9. Figure 5. The data are very interesting and present a nice analysis of the relationship between Adam17 and Cux1 expression.

We thank the referee for this positive feedback and appreciation of our findings

10. Figure 6. This figure is confusing since it presents data in breast cancer, not prostate cancer. The *Ptenpc*^{-/-}; *Trp53pc*^{-/-} mouse model might have been more useful, especially for panels d-f.

We thank the referee for their suggestion and we have now moved the data on breast cancer from Fig. 6 to the Supplementary Fig. 7. We still believe that the data regarding CUX-1 overexpression in breast cancer, *in vivo* (transgenic mouse model) are important to further strengthen the relevance of our findings

11. A general source of difficulty in evaluating this study is the use of numerous prostate cell lines without providing any rationale as to why one cell line is used over another. Moreover, it is not clear why the data was never confirmed using a second cell line, which expressed similar levels of the protein of interest.

We apologize with this referee for not being clear regarding this point. In this study we have used four cell lines, namely 22Rv1, DU-145, LNCaP and PC3. These cell lines are among the most commonly used PCa cell lines in the field of prostate cancer research. Moreover 22Rv1 have WT PTEN, DU-145 heterozygous loss of PTEN and both LNCaP and PC3 have loss of two copies of PTEN. Therefore we used these cells to show that either partial or complete loss of PTEN can promote activation of NOTCH1 by upregulating ADAM17. In the new version of the text we have included a better description of the cell lines used for the experiments to clarify this point.

References:

- 1 Hancox, U. *et al.* Inhibition of PI3Kbeta signaling with AZD8186 inhibits growth of PTEN-deficient breast and prostate tumors alone and in combination with docetaxel. *Mol Cancer Ther* **14**, 48-58, doi:10.1158/1535-7163.MCT-14-0406 (2015).
- 2 Yashiro-Ohtani, Y. *et al.* Pre-TCR signaling inactivates Notch1 transcription by antagonizing E2A. *Genes Dev* **23**, 1665-1676, doi:10.1101/gad.1793709 (2009).
- 3 Weng, A. P. *et al.* c-Myc is an important direct target of Notch1 in T-cell acute lymphoblastic leukemia/lymphoma. *Genes Dev* **20**, 2096-2109, doi:10.1101/gad.1450406 (2006).
- 4 Girard, L. *et al.* Frequent provirus insertional mutagenesis of Notch1 in thymomas of MMTVD/myc transgenic mice suggests a collaboration of c-myc and Notch1 for oncogenesis. *Genes Dev* **10**, 1930-1944 (1996).
- 5 Robey, E. *et al.* An activated form of Notch influences the choice between CD4 and CD8 T cell lineages. *Cell* **87**, 483-492 (1996).
- 6 Li, J. *et al.* PTEN, a putative protein tyrosine phosphatase gene mutated in human brain, breast, and prostate cancer. *Science* **275**, 1943-1947 (1997).
- 7 Alimonti, A. *et al.* Subtle variations in Pten dose determine cancer susceptibility. *Nat Genet* **42**, 454-U136, doi:10.1038/ng.556 (2010).
- 8 Trotman, L. C. *et al.* Pten dose dictates cancer progression in the prostate. *Plos Biol* **1**, 385-396, doi:ARTN e59
10.1371/journal.pbio.0000059 (2003).
- 9 Chen, Z. B. *et al.* Crucial role of p53-dependent cellular senescence in suppression of Pten-deficient tumorigenesis. *Nature* **436**, 725-730, doi:10.1038/nature03918 (2005).
- 10 Zhang, C. C. *et al.* Biomarker and pharmacologic evaluation of the gamma-secretase inhibitor PF-03084014 in breast cancer models. *Clin Cancer Res* **18**, 5008-5019, doi:10.1158/1078-0432.ccr-12-1379 (2012).
- 11 Wei, P. *et al.* Evaluation of selective gamma-secretase inhibitor PF-03084014 for its antitumor efficacy and gastrointestinal safety to guide optimal clinical trial design. *Mol Cancer Ther* **9**, 1618-1628, doi:10.1158/1535-7163.mct-10-0034 (2010).
- 12 Zhang, C. C. *et al.* Synergistic effect of the gamma-secretase inhibitor PF-03084014 and docetaxel in breast cancer models. *Stem Cells Transl Med* **2**, 233-242, doi:10.5966/sctm.2012-0096 (2013).
- 13 Samon, J. B. *et al.* Preclinical analysis of the gamma-secretase inhibitor PF-03084014 in combination with glucocorticoids in T-cell acute lymphoblastic leukemia. *Mol Cancer Ther* **11**, 1565-1575, doi:10.1158/1535-7163.MCT-11-0938 (2012).

Reviewers' comments:

Reviewer #1 (Remarks to the Author):

The revised manuscript satisfactorily addresses my concerns. The findings presented are novel and of potentially great significance from a cancer biology standpoint as well as from a clinical standpoint.

Two minor points remain to be addressed:

- 1) Page 6, line 126: "a strong increase" (correct typo)
- 2) Discussion, Page 13, lines 283-286: The figure presented in the Rebuttal Letter as Rebuttal Figure 1 should be shown as a Supplemental Figure, as the authors suggest in their Rebuttal Letter. This is because the abundance of NOTCH1 transcript itself may become a useful efficacy biomarker for Notch inhibitors

Reviewer #2 (Remarks to the Author):

General Comments:

Based on the reviewers' comments, the authors have significantly revised the manuscript. This has enhanced the logical flow of the manuscript and strengthened the data, the analysis of their findings, and the subsequent conclusions.

Specific comments:

1. It might be helpful to proof-read the revisions again. There are a few words which may not convey the authors' thoughts. For example, manuscript page 3, line 68, the word "pertaining". Should this read "exhibiting"?
2. Regarding the authors' response to Comments #5 and #6, there are a few remaining concerns:
 - a. The authors refer to a previously published paper from their group which has demonstrated that high-grade pre-tumoral PTEN^{-/-} lesions progress to invasive prostate cancer in 15 week old mice (reference 9). While this may be true in the published study, the current study does not appear to demonstrate an invasive phenotype in any of the histopathology shown, including Figure 1, Figure 3a, and Supplementary Figure 3.

Unfortunately, the criteria for determining cell invasion have not been rigorously defined. Localized invasion, that is invasion of prostate cancer cells within the prostate organ, needs to be critically analyzed as follows. If there is a suspect lesion, the pathologist will analyze a number of serial sections before and after the putative invasive cells to ensure that these cells are not a tangential cross-section of a prostatic gland. Because the glandular structure of the prostate is highly convoluted, most of these putative invading cells are found to be attached to a prostatic gland through the analysis of serial sections. This appears to be the case in Figure 3a where the white arrow points to a part of glandular structure that is still attached to the main gland.

The standard clinical criteria for determining whether cancer cells are invading the prostatic stroma are as follows. Putative invading cells should be clearly separated from the prostatic gland. There should be a defined layer of stromal cells between the gland and the invading cells. Furthermore, the thicker the layer of stromal cells separating a cluster of cells from the main gland, the more probable it is that these are invading cells. Using a stromal-specific stain would assist in identifying invading cells. The study conducted by Hong Wu (Cancer Cell 2003, 4:209-221) reports invasive adenocarcinoma in Pten null prostates in Figure 5, red arrows. Perhaps this figure might be helpful in identifying invasive cells in mouse prostates in the present study.

b. The authors state that the PTEN model "fully recapitulating the natural history of human prostate cancer". This statement is a concern since it is generally believed that mouse models do not recapitulate the full spectrum of human prostate cancer. This conclusion has been definitively discussed in many reviews, including a comprehensive review conducted by the Prostate Pathology Committee, Mouse Models of Human Cancer Consortium, National Cancer Institute (Cancer Research 2004, 64:2270-2305). Therefore, the statement should be modified accordingly.

c. The authors state that "the preclinical trial of γ -secretase inhibitor PF-03084014 was mainly performed to determine the anti-tumor efficacy in treating PTEN-deficient prostate cancer patients and it was not our intention to propose this treatment as a preventive agent in healthy subjects". This statement is confusing. A fundamental reason as to why healthy wild-type and single gene deletion control animals are added to a pre-clinical trial using a mouse model is to ensure that the putative anti-tumor efficacy of PF-03084014 in decreasing tumor growth, etc. is "real" and not just a general side-effect of the drug. Indeed, a lack of side-effects on normal tissues would strengthen the outcomes of the pre-clinical trial and these observations should be included in the discussion.

Reviewer #1:

We thank the referee for his/her comments/suggestions that have helped us to improve the overall quality of our manuscript. We have revised our manuscript incorporating all the changes suggested by the referee.

1) Page 6, line 126: "a strong increase" (correct typo)

We apologize for this typo and we have corrected this mistake in our revised version of the manuscript as highlighted in red.

2) Discussion, Page 13, lines 283-286: The figure presented in the Rebuttal Letter as Rebuttal Figure 1 should be shown as a Supplemental Figure, as the authors suggest in their Rebuttal Letter. This is because the abundance of NOTCH1 transcript itself may become a useful efficacy biomarker for Notch inhibitors

We thank the referee for this suggestion. In the new version of the manuscript we have now included this data as Supplementary Fig.2e. We have also included the description of this figure in the text of the manuscript (see text in red).

Reviewer #2:

We thank the referee for his/her constructive comments and critique. We believe that these suggestions and changes, that we have now incorporated in our manuscript, have enhanced the quality of our manuscript. Below we have addressed in full all the comments/concerned raised by the referee.

1. It might be helpful to proof-read the revisions again. There are a few words which may not convey the authors' thoughts. For example, manuscript page 3, line 68, the word "pertaining". Should this read "exhibiting"?

We thank the referee for the suggestions. We have proof-read our revised manuscript including the changes suggested by the referee as highlighted in red.

2. Regarding the authors' response to Comments #5 and #6, there are a few remaining concerns:

a. The authors refer to a previously published paper from their group which has demonstrated that high-grade pre-tumoral PTEN^{-/-} lesions progress to invasive prostate cancer in 15 week old mice (reference 9). While this may be true in the published study, the current study does not appear to demonstrate an invasive phenotype in any of the histopathology shown, including Figure 1, Figure 3a, and Supplementary Figure 3.

Unfortunately, the criteria for determining cell invasion have not been rigorously defined. Localized invasion, that is invasion of prostate cancer cells within the prostate organ, needs to be critically analyzed as follows. If there is a suspect lesion, the pathologist will analyze a number of serial sections before and after the putative invasive cells to ensure that these cells are not a tangential cross-section of a prostatic gland. Because the glandular structure of the prostate is highly convoluted, most of these putative invading cells are found to be attached to a prostatic gland

through the analysis of serial sections. This appears to be the case in Figure 3a where the white arrow points to a part of glandular structure that is still attached to the main gland.

The standard clinical criteria for determining whether cancer cells are invading the prostatic stroma are as follows. Putative invading cells should be clearly separated from the prostatic gland. There should be a defined layer of stromal cells between the gland and the invading cells. Furthermore, the thicker the layer of stromal cells separating a cluster of cells from the main gland, the more probable it is that these are invading cells. Using a stromal-specific stain would assist in identifying invading cells. The study conducted by Hong Wu (Cancer Cell 2003, 4:209-221) reports invasive adenocarcinoma in Pten null prostates in Figure 5, red arrows. Perhaps this figure might be helpful in identifying invasive cells in mouse prostates in the present study.

We thank the referee for this comment and we also apologize for not being clear about

these figures. In our opinion, there is no doubt that the γ -secretase inhibitor tested in our models is effective in blocking tumorigenesis as demonstrated by the general appearance of the tumor lesions (less glands with tumors) reduced tumor size, decreased Ki-67 staining and decreased stroma

reaction observed in several figures such as for instance Figure 2. Regarding the specific figures mentioned by this referee: 1) We would like to clarify that the image of *Pten*^{pc/-} tumors in Fig.1 does not display invasive phenotype because these tumors were resected from 12-weeks old *Pten*^{pc/-} mice that as previously reported develop HG-PIN and not invasive adenocarcinoma¹. We have therefore included this clarification in the figure legend of the new version of the manuscript (see text in red). 2) We agree with this referee that the previous Figure 3 and Supplementary Figure 3b of the manuscript do not faithfully represent the phenotype of focal invasive prostate tumors. Therefore, we have changed these two figures to better represent cases of invasive prostate cancer. In these two new figures, invading cluster of tumor cells are clearly separated from the prostatic tumor gland by a defined layer of thick stromal cells. Moreover, as also mention by the referee, the analysis of serial H&E sections of *Pten*^{pc/-}; *Trp53*^{pc/-} tumor samples (see Figure 1 of the rebuttal) show that stroma-infiltrating tumor cells that appear in the earlier section of Case 1 and 2 disappear in the subsequent sections (see black and red arrows in Figure 1 of the rebuttal). As requested by the editor, we have also performed immunofluorescence stainings in these tumor sections for Vimentin (Stroma-specific

staining)/ E-cadherin (Epithelial-specific staining) (see new Figure 3, Supplementary Figure 3b and Figure 2 of this rebuttal). These stainings confirm the presence of isolated epithelial-tumor cells, clearly separated from the gland by a thick stroma layer. According to our pathologist, these images are in line with the one reported in the Hong Wu paper (Cancer Cell 2003, 4:209-221). However in the case cited by the referee the phenotype looks much more aggressive than in our tumors, probably due to the fact that the Hong Wu mice were castrated at an early time point and analysed long time after castration (selection of aggressive tumor clones?). We believe that these stainings should now satisfy the concerns raised by the referee.

b. The authors state that the PTEN model "fully recapitulating the natural history of human prostate cancer". This statement is a concern since it is generally believed that mouse models do not recapitulate the full spectrum of human prostate cancer. This conclusion has been definitively discussed in many reviews, including a comprehensive review conducted by the Prostate Pathology Committee, Mouse Models of Human Cancer Consortium, National Cancer Institute (Cancer Research 2004, 64:2270-2305). Therefore, the statement should be modified accordingly.

We thank the referee for this clarification. However, this sentence does not appear in the main text of the manuscript.

c. The authors state that "the preclinical trial of α -secretase inhibitor PF-03084014 was mainly performed to determine the anti-tumor efficacy in treating PTEN-

deficient prostate cancer patients and it was not our intention to propose this treatment as a preventive agent in healthy subjects". This statement is confusing. A fundamental reason as to why healthy wild-type and single gene deletion control animals are added to a pre-clinical trial using a mouse model is to ensure that the putative anti-tumor efficacy of PF-03084014 in decreasing tumor growth, etc. is "real" and not just a general side-effect of the drug. Indeed, a lack of side-effects on normal tissues would strengthen the outcomes of the pre-clinical trial and these observations should be included in the discussion.

We thank the referee for this clarification and we are happy that he is now satisfy with our findings.

References:

- 1 Chen, Z. *et al.* Crucial role of p53-dependent cellular senescence in suppression of Pten-deficient tumorigenesis. *Nature* **436**, 725-730, doi:10.1038/nature03918 (2005).

REVIEWERS' COMMENTS:

Reviewer #2 (Remarks to the Author):

Response to Previous Comments

1. The immunohistochemistry has been improved considerably. Hopefully this more rigorous staining approach has been taken into consideration and used to re-evaluate the previous data to ensure that the first analysis was accurate and representative of prostate cancer progression in the Pten mouse model.

2. Regarding the reviewers comments to Comment 2b, "We thank the referee for this clarification. However, this sentence does not appear in the main text of the manuscript".

Please see the Discussion section, page 13, lines 282- 283 where the authors emphatically state, "To address this question, we used Ptenpc^{-/-} and Ptenpc^{-/-}; Trp53pc^{-/-} mouse models which fully recapitulate different stages of human PCa."

This statement should be removed. It is an incorrect statement that perpetuates the commonly held myth (i.e., a widely held but false belief) that mouse prostate cancer is the same as human prostate cancer. It has been clearly demonstrated in the literature that none of the mouse models generated to date fully recapitulate the development, progression, and metastatic spread of human PCa.

Additional Comments

1. Most of the following figure panels have error bars but do not appear to be evaluated statistically. Please provide statistical evaluation (i.e., p values) for the following figures:
Figure 1, panel (i),
Figure 4, panels (e) and (k),
Figure 5, panels (b) and (j),
Supplementary Fig. 1, panels (a), (e), and (g),
Supplementary Fig. 2, panels (a) and (e),
Supplementary Fig. 3, panels (a) and (b),
Supplementary Fig. 4, panels (b) and (d), and
Supplementary Fig.7, panels (b) and (f).

We would also like to thank the referee 2 for his/her suggestions and critique regarding our manuscript. Based on the comments/suggestions from referee 2, we have now revised our manuscript incorporating all the required changes and missing detailed evaluation of our data that has helped to improve the overall quality of the manuscript. We thank the referee for his/her consideration of our rigorous stainings and quantification of invasiveness of *Pten*^{pc/-} and *Pten*^{pc/-}; *Trp53*^{pc/-} tumours.

Referee 2 comments:

1. The immunohistochemistry has been improved considerably. Hopefully this more rigorous staining approach has been taken into consideration and used to re-evaluate the previous data to ensure that the first analysis was accurate and representative of prostate cancer progression in the *Pten* mouse model.

We thank the referee for his/her consideration of our stainings. We have incorporated these stainings including the statistical evaluation in our manuscript in Fig. 3a and Supplementary Fig. 3a and b.

2. Regarding the reviewers comments to Comment 2b, “We thank the referee for this clarification. However, this sentence does not appear in the main text of the manuscript”.

Please see the Discussion section, page 13, lines 282- 283 where the authors emphatically state, “To address this question, we used *Pten*^{pc/-} and *Pten*^{pc/-}; *Trp53*^{pc/-} mouse models which fully recapitulate different stages of human PCa.”

This statement should be removed. It is an incorrect statement that perpetuates the commonly held myth (i.e., a widely held but false belief) that mouse prostate cancer is the same as human prostate cancer. It has been clearly demonstrated in the literature that none of the mouse models generated to date fully recapitulate the development, progression, and metastatic spread of human PCa.

We apologize to the referee for the statement that remained in the “Discussion” part of our manuscript text. In our revised manuscript we have now removed this statement from the Discussion.

Additional Comments

1. Most of the following figure panels have error bars but do not appear to be evaluated statistically. Please provide statistical evaluation (i.e., p values) for the following figures:

Figure 1, panel (i),

Figure 4, panels (e) and (k),

Figure 5, panels (b) and (j),

Supplementary Fig. 1, panels (a), (e), and (g),

Supplementary Fig. 2, panels (a) and (e),

Supplementary Fig. 3, panels (a) and (b),

Supplementary Fig. 4, panels (b) and (d), and

Supplementary Fig.7, panels (b) and (f).

We apologize to the referee for missing out the statistical evaluation of our data. In the revised version of manuscript we have incorporated all the statistical analysis wherever missing. We also thank the referee for his/her valuable observation regarding this point.